# DEEP EXPLORATION WITH PAC-BAYES

## ABSTRACT

Reinforcement learning for continuous control under sparse rewards is an under-explored problem despite its significance in real life. Many complex skills build on intermediate ones as prerequisites. For instance, a humanoid locomotor has to learn how to stand before it can learn to walk. To cope with reward sparsity, a reinforcement learning agent has to perform deep exploration. However, existing deep exploration methods are designed for small discrete action spaces, and their successful generalization to state-of-the-art continuous control remains unproven. We address the deep exploration problem for the first time from a PAC-Bayesian perspective in the context of actor-critic learning. To do this, we quantify the error of the Bellman operator through a PAC-Bayes bound, where a bootstrapped ensemble of critic networks represents the posterior distribution, and their targets serve as a data-informed function-space prior. We derive an objective function from this bound and use it to train the critic ensemble. Each critic trains an individual actor network, implemented as a shared trunk and critic-specific heads. The agent performs deep exploration by acting deterministically on a randomly chosen actor head. Our proposed algorithm, named *PAC-Bayesian Actor-Critic (PBAC)*, is the only algorithm to successfully discover sparse rewards on a diverse set of continuous control tasks with varying difficulty.

## 1 INTRODUCTION

Complex control tasks commonly presuppose the completion of a sequence of sub-tasks. For example, winning a chess game involves following a theoretically correct opening, a middle game with a chain of creative tactical combinations, and an error-free endgame. Similarly, in Montezuma's Revenge, an agent must visit multiple rooms and collect keys in the correct order. A genuinely intelligent agent is expected to identify and solve these sub-tasks based solely on the final reward. Deep reinforcement learning (RL) has been highly successful in handling such sparse reward scenarios. AlphaZero (Silver et al., 2018) and Random Network Distillation (RND) (Burda et al., 2019) reach super-human level performance scores in chess and Montezuma's Revenge, respectively, despite receiving zero reward for long interaction sequences.

Continuous control tasks share a similar modular structure. There is strong evidence that biological systems execute complex movements through robust and stable building blocks, known as motion primitives (Flash & Hochner, 2005). Application of the same idea to robotics, called dynamic movement primitives (Schaal, 2006), demonstrated remarkable success when unit movement patterns are fit to observations separately and an agent is trained to apply them in a sequence. This recipe works well when the target environment is well-understood and eligible for a rigorous engineering effort. However, performing loosely defined tasks in open-world scenarios requires tabula rasa approaches that both invent these primitives from authentic environment interactions and use them in an optimal order for task completion within feasible sample complexity limits. Reinforcement learning has not yet delivered success stories in continuous control setups with sparse rewards.

The common ground of reinforcement learning approaches developed for sparse reward setups is to quantify the uncertainty on the Bellman target and use it for exploration. The exploration scheme should aim for multi-step information gains by propagating uncertainty across Bellman backups, a notion known as *deep exploration* (Osband et al., 2016a). There exist several model-free approaches that treat deep exploration as a Bayesian inference problem on the Bellman target (Osband et al., 2016a; 2018; Touati et al., 2020; Fellows et al., 2024). Additionally, model-based approaches explore learning a distribution over the state transition kernel and propagating its uncertainty through

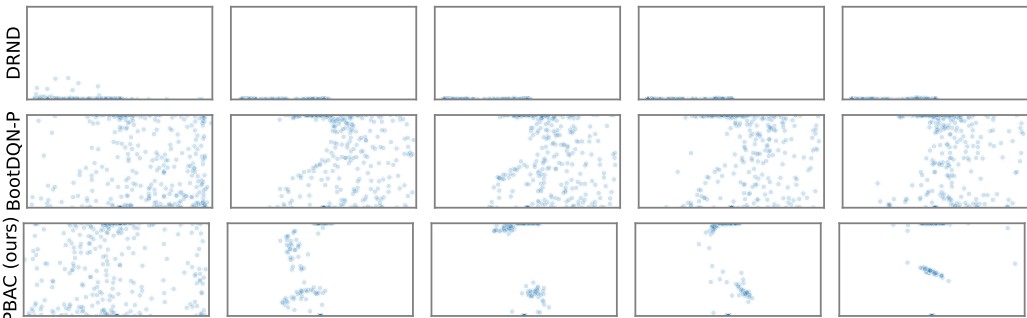

Figure 1: *Deep exploration.* State exploration patterns of the state and angle dimension in the cartpole environment are visualized as training progresses (every 500th state throughout the training is recorded and visualized in five groups, left to right). After an initial phase of broad exploration, PBAC quickly discovers the reward region (top middle in each of the scatter plots) and learns to exploit it while continuing to explore to a small amount. DRND (Yang et al., 2024) and BootDQN-P (Osband et al., 2018) on the other hand struggle with this task. DRND gets stuck early on, while BootDQN-P manages to find the reward region but continues to explore excessively, preventing full exploitation. See the experimental section (Section 4) for more details.

the value function to estimate the Bellman target uncertainty (O'Donoghue et al., 2018; Luis et al., 2023b). Finally, there are pseudo-count-based approaches in the middle ground of model-based and model-free approaches that aim to estimate the visitation count of state-action pairs by distilling a randomly initialized predictor into a parametric auxiliary network (Nikulin et al., 2023; Yang et al., 2024; Burda et al., 2019). Among all these approaches, only a few recent model-based approaches (Luis et al., 2023b) show success in continuous control tasks at the expense of a prohibitive computational overhead for model learning and a strong dependency on learnable environment dynamics, which would essentially make model predictive control sufficient for many real-world applications.

**Contribution.** We formulate the deep exploration problem for the first time from a Probably Approximately Correct (PAC) Bayesian (McAllester, 1999; Alquier et al., 2024) perspective and develop an actor-critic algorithm that reaches unprecedented performance in continuous control with sparse rewards. To do this, we quantify the Bellman operator error using a generic PAC-Bayes bound formulation (Germain et al., 2009) for the first time in a policy evaluation context. We treat a bootstrapped ensemble of critic networks as an empirical posterior distribution and build a data-informed function-space prior from their corresponding target networks. To train this ensemble, we derive a simple and intuitive objective function. As the actor, we use a deterministic network with a shared trunk and multiple heads, each assigned to a separate critic. We apply posterior sampling during training time for exploration and Bayesian model averaging during evaluation time. Finally, we conduct experiments on a diverse set of continuous control tasks, both established and novel, some of which have reward structures significantly more challenging than the prior art. Our *PAC-Bayesian Actor-Critic (PBAC)* algorithm is the only model capable of solving these tasks, whereas both state-of-the-art and well-established methods fail in several. For example, Figure 1 visualizes PBAC's deep exploration followed by effective exploitation, in contrast to two baselines.

## 2 BACKGROUND AND PRIOR ART

### 2.1 REINFORCEMENT LEARNING

Consider a measurable space $(\mathcal{S}, \sigma(\mathcal{S}))$ of a set of states $\mathcal{S}$ an agent may be in and its corresponding $\sigma$-algebra $\sigma(\mathcal{S})$ comprising all measurable subsets of the state space $\mathcal{S}$ and an action space $\mathcal{A}$ from which an agent can choose an action to interact with its environment. We define a Markov Decision Process (MDP) (Puterman, 2014) as a tuple $M = \langle \mathcal{S}, \mathcal{A}, r, P, P_0, \gamma \rangle$, where $r : \mathcal{S} \times \mathcal{A} \to [0, R]$ is a bounded reward function, $P : \mathcal{S} \times \mathcal{A} \times \sigma(\mathcal{S}) \to [0, 1]$ is a state transition kernel conditioned on a state-action pair, meaning that $P(\cdot|s, a)$ is the probability distribution of a next state $s' \in \mathcal{S}$ when starting from a current state-action pair $(s, a) \in \mathcal{S} \times \mathcal{A}$. $P_0 : \sigma(\mathcal{S}) \to [0, 1]$ is an initial state

distribution, and $\gamma \in (0, 1)$ is a discount factor. We denote a policy as a map $\pi : \mathcal{S} \to \mathcal{A}$. Reinforcement learning aims to learn a policy that maximizes the expected sum of discounted rewards, $\pi' := \arg\max_{\pi \in \Pi} \mathbb{E}_{\tau_\pi} \left[ \sum_{t=0}^{\infty} \gamma^t r(s_t) \right]$ where the expectation is taken with respect to the trajectory $\tau_\pi := (s_0, a_0, s_1, a_1, s_2, a_2, \ldots)$ of states and actions that occur when a policy $\pi$ chosen from a feasible set $\Pi$ is employed. The exact Bellman operator for a policy $\pi$ is defined as

$$T_\pi Q(s, a) = r(s, a) + \gamma \mathbb{E}_{s' \sim P(\cdot|s,a)} \left[ Q(s', \pi(s')) \right],$$

for some $Q : \mathcal{S} \times \mathcal{A} \to \mathbb{R}$. The unique fixed point of this operator is the true action-value function $Q_\pi$ that maps a state-action pair $(s, a)$ to the discounted reward a policy $\pi$ collects when executed starting from $(s, a)$, that is $T_\pi Q(s, a) = Q(s, a)$ if and only if $Q(s, a) = Q_\pi(s, a), \forall(s, a)$. Any other $Q$ incurs an error $T_\pi Q(s, a) - Q(s, a)$ referred to as the *Bellman error*. Define the squared $P_\pi$-weighted $L_2$-norm as $\|f\|_{P_\pi}^2 = \int_{s \in \mathcal{S}} |f(s)|^2 dP_\pi(s)$ for some $f : \mathcal{S} \to \mathbb{R}$. We aim to approximate the true action-value function $Q_\pi$ by one-step Temporal Difference (TD) learning that minimizes

$$L(Q) := \|T_\pi Q - Q\|_{P_\pi}^2 \tag{1}$$

with respect to $Q$ given a data set $\mathcal{D}$. In customary setups, the transition kernel is not known, hence only a stochastic version of this loss can be calculated via Monte Carlo samples

$$\widetilde{L}(Q) := \mathbb{E}_{s \sim P_\pi} \left[ \mathbb{E}_{s' \sim P(\cdot|s, \pi(s))} \left[ (\widetilde{T}_\pi Q(s, \pi(s), s') - Q(s, a))^2 \right] \right],$$

with the *sample Bellman operator*, defined as

$$\widetilde{T}_\pi Q(s, a, s') = r(s, a) + \gamma Q(s', \pi(s')).$$

A deep reinforcement learning algorithm fits a function approximator $Q$ to a set of observed tuples $(s, a, s')$ stored in a replay buffer $\mathcal{D}$ by minimizing an empirical estimate of the stochastic loss:

$$\widehat{L}_{\mathcal{D}}(Q) := \frac{1}{|\mathcal{D}|} \sum_{(s,a,s') \in \mathcal{D}} \left( \widetilde{T}_\pi Q(s, a, s') - Q(s, a) \right)^2.$$

If the function family contains the true action-value function, the optimization problem above can be solved in full precision, and the environment is explored with a behavior policy that has full coverage on the state-action space. Temporal difference learning is then guaranteed to converge to the true solution asymptotically (Bertsekas & Tsitsiklis, 1996). Because these conditions are rarely satisfied in practice, reinforcement learning algorithms commonly suffer from sample inefficiency which amounts to prohibitive costs in real-world applications.

## 2.2 DEEP EXPLORATION

The classical optimal control paradigm prescribes a rigorous design of cost functions, i.e., negative rewards, that give a dense signal about model performance at each stage of model fitting (Stengel, 1994; Kirk, 2004). This approach is akin to detailed loss designs in the common practice of deep learning. However, next-generation AI problems demand agents with high cognitive capabilities that can autonomously learn smart strategies to reach distant high rewards without being trapped by intermediate small rewards or prematurely giving up searching. These approaches are commonly known as *deep exploration* (Osband et al., 2016a). Their common ground is to estimate how often a state-action pair has been visited until a certain training step and direct the exploration towards unvisited ones. Deep exploration approaches can be classified into four categories based on how accurate knowledge of environment dynamics they assume to make this estimation:

*(i) Model-based approaches* learn to infer the state transition kernel by a function approximator $P_\phi(\cdot|s, a) \approx P(\cdot|s, a)$ and use its uncertainty to generate *intrinsic rewards* (Chentanez et al., 2004) to perform curiosity-driven exploration, which has remarkable roots to biological intelligence (Schmidhuber, 2010; Modirshanechi et al., 2023). VIME (Houthooft et al., 2016) incentivizes exploration of states that would bring more information gain on the estimated transition kernel, while BYOL-Explore (Guo et al., 2022) and BYOL-Hindsight (Jarrett et al., 2023) choose states with higher prediction error. Approaches such as UBE (O'Donoghue et al., 2018), Exact UBE (Luis et al., 2023b), and QU-SAC (Luis et al., 2023a) choose the target quantity as the sample Bellman operator $\widetilde{T}_\pi Q$. MOBILE (Sun et al., 2023) and MOPO (Yu et al., 2020) are the respective offline RL counterparts of these strategies.

*(ii) Pseudo-count approaches* learn to predict only the relative visitation frequencies of state-action pairs without needing to predict the state transitions. These frequencies are then used to generate intrinsic rewards to direct the exploration towards less frequent state-action pairs. The commonplace approach, called *Random Network Distillation (RND)* (Burda et al., 2019), distills a fixed network with randomly initialized weights into another network with learned parameters. It uses the success of the distillation as a pseudo-count to generate intrinsic rewards. State-of-the-art variants include SAC-RND (Nikulin et al., 2023) and SAC-DRND (Yang et al., 2024).

*(iii) Randomized value iteration approaches* learn to estimate the uncertainty around the approximate $Q$ function by modeling it as a random variable and performing posterior inference based on the observed Bellman errors. Proposed inference methods include Bayesian linear regression (Osband et al., 2016b), mean-field (Lipton et al., 2018) or structured (Touati et al., 2020) variational inference on Bayesian neural networks, Bayesian deep bootstrap ensembles with flat priors (Osband et al., 2016a) and informative priors (Osband et al., 2018). Fellows et al. (2021) find out that these approaches actually infer a posterior on the Bellman target $T_\pi Q$ instead of the intended $Q$ and propose to overcome the consequences of this mismatch by training a separate $Q$ network on the posterior predictive mean of $T_\pi Q$. A more advanced version of this approach, named *Bayesian Exploration Networks (BEN)* (Fellows et al., 2024), fits a Bayesian heteroscedastic neural net on the Bellman target using structured variational inference.

*(iv) Policy randomization approaches* propose different perturbations on the learned models to randomize the employed policy without aiming to quantify the uncertainty of a quantity. Proposed solutions include perturbing the $Q$-network weights for discrete action spaces (Fortunato et al., 2018), actor-network weights (Plappert et al., 2018) for continuous control, and transforming the policy distribution by normalizing flows (Mazoure et al., 2019).

In this work, we follow the randomized value iteration approach due to the balance it grants between computational feasibility and theoretical soundness. Notably, Bayesian model averaging gives the optimal decision rule for an approximate $Q$ function with partial observation on an MDP with reliable uncertainty estimates (Cox & Hinkley, 1974). We approach the deep exploration problem for the first time from a PAC-Bayesian perspective. The use of PAC-Bayes in reinforcement learning is limited to theoretical analysis (Fard et al., 2012) and preliminary developments aiming to improve training robustness (Tasdighi et al., 2024).

## 2.3 PAC-BAYESIAN ANALYSIS AND LEARNING

Given inputs $\mathcal{Z}$, outputs $\mathcal{Y}$, a set of hypotheses $\mathcal{H} = \{h : \mathcal{Z} \rightarrow \mathcal{Y}\}$ that map inputs to outputs, and a loss function $\ell : \mathcal{Y} \times \mathcal{Y} \rightarrow [0, B]$, we define the empirical risk $\widehat{L}$ for a set of observations $\mathcal{O} = \{(z_i, y_i) : (z_i, y_i) \sim P_D, i \in [n]\}$[1] containing $n$ samples obtained from a distribution $P_D$, and the corresponding true risk $L$ as

$$\widehat{L}_\mathcal{O}(h) = \frac{1}{n} \sum_{i=1}^{n} \ell(h(z_i), y_i), \qquad L(h) = \mathbb{E}_{(z,y) \sim P_D} [\ell(h(z), y)].$$

PAC-Bayesian analysis (Shawe-Taylor & Williamson, 1997; Alquier et al., 2024) characterizes an upper bound $C$ of the form $d(\mathbb{E}_{h \sim \rho}[L(h)], \mathbb{E}_{h \sim \rho}[\widehat{L}_\mathcal{O}(h)]) \leq C(\rho, \rho_0, \delta)$ that holds with probability at least $1 - \delta$ for any error tolerance $\delta \in (0, 1]$ and any prior probability measure $\rho_0$ and posterior probability measure $\rho$ chosen from the set of all measures $\mathcal{P}$ feasible on the hypothesis space $\mathcal{H}$ for a given convex distance metric $d(\cdot, \cdot)$. The choice of the posterior measure $\rho$ can depend on the data $\mathcal{O}$ while the prior measure $\rho_0$ cannot, hence their names. However, unlike in Bayesian inference, the posterior and the prior do not need to relate to each other via a likelihood function. The prior serves as a reference with respect to which a generalization statement is made. Various well-known PAC-Bayes bounds (such as McAllester (1999); Seeger (2002); Catoni (2007)) can be shown as instances of the generic form below, the proof of which we present in Appendix A for completeness.

**Theorem 1** (Germain et al. (2009)). *For any convex function $d : [0, B] \times [0, B] \rightarrow \mathbb{R}$, distributions $\rho, \rho_0$ measurable on the hypothesis space $\mathcal{H}$, and error tolerance $\delta \in (0, 1]$, simultaneously, the following inequality holds with probability at least $1 - \delta$:*

$$d\left(\mathbb{E}_{h \sim \rho}\left[\widehat{L}_\mathcal{O}(h)\right], \mathbb{E}_{h \sim \rho}[L(h)]\right) \leq \frac{1}{n}\left(\mathrm{KL}\left(\rho \parallel \rho_0\right) + \ln\left(\frac{1}{\delta}\mathbb{E}_{\mathcal{O} \sim P_D}\mathbb{E}_{h \sim \rho_0} e^{nd(\widehat{L}_\mathcal{O}(h), L(h))}\right)\right).$$

---

[1]Here and below $[x]$ denotes the set $\{1, \ldots, x\}$ for integer $x$.

In the bound, $\text{KL}\left(\rho \parallel \rho_0\right) = \mathbb{E}_{h\sim\rho}\left[\log \rho(h) - \log \rho_0(h)\right]$ stands for the Kullback-Leibler divergence between two probability measures.

Since a PAC-Bayes bound holds for any posterior, a parametric family of posteriors can be chosen and its parameters can be fit to data by minimizing the right-hand side of the bound. This approach, called *PAC-Bayesian Learning*, demonstrated remarkable success in image classification (Dziugaite & Roy, 2017; Wu et al., 2024) and regression (Reeb et al., 2018). There exist only preliminary results such as Tasdighi et al. (2024) that use this approach in deep reinforcement learning. The tightness of the bound is determined by the choice of $d$, as well as additional assumptions and algebraic manipulations on the second term on the right-hand side of the inequality. Remarkably, this term does not depend on $\rho$, and hence does not play any role in a PAC-Bayesian learning algorithm. Our solution instrumentalizes this simple but often overlooked observation.

## 3 Deep exploration with a PAC-Bayesian actor-critic

### 3.1 Theory

We aim to build an actor-critic algorithm able to perform deep exploration in continuous control setups with sparse rewards. We consider a model-free and off-policy approximate temporal difference training setup. The only existing prior work at the time of writing that addresses this setup with a PAC-Bayesian analysis is by Fard et al. (2012). They use PAC-Bayes to directly bound the value of a policy from a single chain of observations. The Markovian dependencies of these observations necessitate building on a Bernstein-like concentration inequality that works only for extremely long episodes, prohibiting its applicability to PAC-Bayesian learning (Tasdighi et al., 2024). We overcome this limitation by instead developing a PAC-Bayes bound on the Bellman operator error. See Appendix A for proofs on all results presented in this section.

We assume the existence of a replay buffer $\mathcal{D}$ containing samples obtained from real environment interactions. Following Fard et al. (2012), we assume for convenience that the policy $\pi$ being evaluated always induces a stationary state transition kernel, that is $P_\pi(s') = \int P(s'|s, \pi(s)) P_\pi(s) ds$ for every $s \in \mathcal{S}$ in all training-time environment interactions, where $P_\pi$ denotes the state visitation distribution for policy $\pi$. This is essentially a hidden assumption made by the commonplace off-policy deep reinforcement learning algorithms that take uniformly random batches from their replay buffers. The difference between the sample Bellman operator and the Bellman operator, $\widetilde{T}_\pi Q(s, a, s') - T_\pi Q(s, \pi(s))$, is known to tend to an accumulating positive value when $Q$ is concurrently used as a training objective for policy improvement, leading to the phenomenon known as the *overestimation bias* (Thrun & Schwartz, 1993). The problem is tackled by approaches such as double $Q$-learning (Van Hasselt, 2010) and min-clipping (Fujimoto et al., 2018). We define with

$$X(s, a) \stackrel{d}{=} Q(s, a) + \widetilde{T}_\pi Q(s, a, s') - T_\pi Q(s, a), \qquad \forall (s, a) \sim \mathcal{S} \times \mathcal{A}$$

a new random variable, where $\stackrel{d}{=}$ denotes equality in distribution. This is a hypothetical variable that predicts from $(s, a)$ the value of the observed Bellman target caused by the randomness of $s'$. It is related to the action-value function approximator as follows

$$Q(s, a) = Q(s, a) + \mathbb{E}_{s'\sim P(\cdot|s,a)}\left[\widetilde{T}_\pi Q(s, a, s')\right] - T_\pi Q(s, a)$$

$$= \mathbb{E}_{s'\sim P(\cdot|s,a)}\left[Q(s, a) + \widetilde{T}_\pi Q(s, a, s') - T_\pi Q(s, a)\right] = \mathbb{E}\left[X(s, a)\right].$$

Let $\rho$ denote the distribution of $X$, then the one-step TD learning loss can be expressed as

$$L(Q) = ||T_\pi Q - Q||^2_{P_\pi} = ||T_\pi Q - \mathbb{E}\left[X\right]||^2_{P_\pi}$$

$$= \mathbb{E}_{s\sim P_\pi}\left[\left(\mathbb{E}_{s'\sim P(\cdot|s,\pi(s))}\left[r(s, \pi(s)) + \gamma\mathbb{E}\left[X(s', \pi(s'))\right]\right] - \mathbb{E}\left[X(s, \pi(s))\right]\right)^2\right]$$

with the corresponding stochastic variant[2]

$$\widetilde{L}(\rho) = \mathbb{E}_{s\sim P_\pi}\mathbb{E}_{s'\sim P(\cdot|s,\pi(s))}\mathbb{E}_{X\sim\rho}\left[\left(\widetilde{T}_\pi X\left(s, \pi(s), s'\right) - X(s, \pi(s))\right)^2\right].$$

---

[2] A single realization of $X$ shares the same domain and range as $Q$. Hence it can be passed through the sample Bellman operator $\widetilde{T}$.

This optimization problem no longer fits a deterministic map $Q$ but instead infers a distribution $\rho$ that best describes the action-value function perturbed by the Bellman operator error. The loss can be interpreted as the true risk of a Gibbs predictor $X \sim \rho$. The corresponding empirical risk is

$$\widehat{L}_{\mathcal{D}}(\rho) = \frac{1}{|\mathcal{D}|} \sum_{(s,a,s') \in \mathcal{D}} \mathbb{E}_{X \sim \rho} \left[ \left( \widetilde{T}_\pi X(s,a,s') - X(s,\pi(s)) \right)^2 \right].$$

We formulate our learning problem as choosing $\rho$ from a feasible set $\mathcal{P}$ that gives the tightest PAC-Bayes bound on $\widetilde{L}(\rho)$ with respect to data set $\mathcal{D}$ and a prior $\rho_0 \in \mathcal{P}$ on the distribution of $X$. This bound should give generalization guarantees about how well $X$ can predict a noisy Bellman operator output. However, in a reinforcement learning setup, our final quantity of interest is the value estimation error $\|Q_\pi - Q\|_{P_\pi} = \|Q_\pi - \mathbb{E}[X]\|_{P_\pi}$, upper bounded via Jensen's inequality by $\mathbb{E}_{X \sim \rho} \|Q_\pi - X\|_{P_\pi}$. Prior work (Fellows et al., 2021) addresses the mismatch between the target and inferred quantities of interest in probabilistic reinforcement learning and its consequences. We propose a different solution by establishing the link using the contraction property of Bellman operators and the following result.

**Lemma 1.** *For any $\rho$ defined on $X$, the following identity holds*

$$\widetilde{L}(\rho) = \mathbb{E}_{X \sim \rho} \|T_\pi X - X\|_{P_\pi}^2 + \gamma^2 \mathbb{E}_{X \sim \rho} [\mathrm{Var}_{s \sim P_\pi}[X(s, \pi(s))]].$$

Applying the contraction property of Bellman operators to $\|T_\pi X - X\|_{P_\pi}^2$ (see Lemma 2) and plugging all quantities into the generic PAC-Bayes bound presented in Theorem 1, we arrive at

**Theorem 2.** *For any posterior and prior measures $\rho, \rho_0 \in \mathcal{P}$, error tolerance $\delta \in (0, 1]$, and deterministic policy $\pi$, simultaneously, the following inequality holds with probability at least $1 - \delta$:*

$$\mathbb{E}_{X \sim \rho} \|Q_\pi - X\|_{P_\pi}^2 \tag{2}$$

$$\leq \frac{1}{(1-\gamma)^2} \left( \widehat{L}_{\mathcal{D}}(\rho) + \frac{1}{n} \left[ KL(\rho \| \rho_0) + \ln \frac{1}{\delta} + \frac{nR^2}{(1-\gamma)^2} \right] - \gamma^2 \mathbb{E}_{X \sim \rho} \mathrm{Var}_{s \sim P_\pi}[X(s, \pi(s))] \right).$$

This bound is vacuous due to the term $(nR^2)/(1-\gamma)^2$. However, it is free from the limitations specialized bounds introduce, for instance, the boundedness of the loss to the unit interval (McAllester, 2003; Seeger, 2002) and the choice of a multiplier on the empirical risk term (Catoni, 2007). Our main concern is to develop an algorithm using learning-theoretic tools and not to provide tight performance guarantees. However, tighter guarantees can be given by plugging the learned posterior of our algorithm into an existing bound rigorously developed for a specific purpose.

### 3.2 Implementation

We take the following steps to implement the bound:

**1)** We model the posterior distribution as an empirical distribution comprising an ensemble of $K$ critic networks, $\mathcal{X} := \{X_k : \mathcal{S} \times \mathcal{A} \to \mathbb{R}, k \in [K]\}$, with corresponding weights $\theta_k$. That is, we have $\rho(A) := \frac{1}{K} \sum_{k=1}^{K} \delta_{X_k}(A)$ for $A \in \sigma(\mathcal{S})$ and Dirac measure $\delta$. We also introduce a target copy, $\bar{X}_k$, for each ensemble element with parameters updated by Polyak averaging as $\bar{\theta}_k \leftarrow (1-\tau)\bar{\theta}_k + \tau\theta_k$ for some $\tau \in [0, 1]$. The resulting empirical risk term for a single observation is $\frac{1}{K} \sum_{k=1}^{K} \left( r + \gamma \bar{X}_k(s', \pi(s')) - X_k(s, a) \right)^2$ where $\bar{X}_k$ are used in the target computations to ensure robust training (Lillicrap et al., 2016).

**2)** We model the distributions $\rho, \rho_0$ directly on the function space due to multiple reasons. Deep reinforcement learning setups use action-value functions in downstream tasks during training, e.g., exploration, bootstrapping, and actor training. Design choices such as how much to explore and how conservatively to evaluate the Bellman target can only be made on the function space. Furthermore, according to the data processing inequality (Polyanskiy & Wu, 2014, Theorem 6.2), the function-space view yields a tighter generalization bound than the weight-space view. The KL divergence between two measures $\rho, \rho_0$ defined on $\mathcal{H}$ can be expressed as (Rudner et al., 2022):

$$\mathrm{KL}(\rho \| \rho_0) = \sup_{\mathcal{D} \in \mathcal{B}} \mathbb{E}_{s \in \mathcal{D}} \left[ \mathbb{E}_{X \sim \rho} \left[ \log f_\rho(X|s, \pi(s)) - \log f_{\rho_0}(X|s, \pi(s)) \right] \right],$$

where $f_\rho$ and $f_{\rho_0}$ are the probability density functions of the two measures evaluated at $X$ for a given state-action pair and $\mathcal{B}$ is the space of all possible data sets $\mathcal{D}$. As the sup calculation is intractable in practice, Rudner et al. (2022) approximate it with

$$\sup_{B \in \{B_j : j \in [J]\}} \frac{1}{J} \sum_{j=1}^{|B_j|} \log f_\rho(s, \pi(s)) - \log f_{\rho_0}(s, \pi(s)),$$

where $B_j$ are minibatches sampled from the available data set. It underestimates the true value at a degree inversely proportional to $J$. We introduce a simpler approximation that is more accurate for any fixed sample: $\sum_{j=1}^{J} \sum_{j=1}^{|B_j|} \log f_\rho(s, \pi(s)) - \log f_{\rho_0}(s, \pi(s))$ since the sum of a set of scalars upper bounds their supremum. We are not aware of any earlier application of function-space priors in the context of PAC-Bayesian learning, especially with application to reinforcement learning.

**3)** Inspired by prior work on PAC Bayes outside the reinforcement learning context (Ambroladze et al., 2006; Dziugaite & Roy, 2018), we adopt a data-informed prior $\rho_0$ to attain a tight PAC Bayes bound that can be used more reliably for policy evaluation. As the most recent and robust information about the action values are stored in the targets $\bar{X}_k$, we use them to build a maximum likelihood estimate of a normal density. Since the target networks are trained to evaluate the next state action-values, we choose $f_{\rho_0}(s', \pi(s')) = \mathcal{N}(\bar{\mu}_\pi(s'), \sigma_0^2)$ where $\bar{\mu}_\pi(s') = \frac{1}{K} \sum_{k=1}^{K} \bar{X}_k(s', \pi(s'))$. Projecting the random variable one-time step backwards, we get $f_{\rho_0}(s, \pi(s)) := \mathcal{N}(r + \gamma \bar{\mu}_\pi(s'), \gamma^2 \bar{\sigma}_0^2)$ where $r$ is the reward observation related to state $s$. Note that the use of critic targets does not violate the assumptions of the PAC-Bayes bound as $\rho_0$ is built on the critic targets evaluated at the previous gradient step and the bound is given only for the current minibatch.

**4)** The posterior $\rho$ defined in Step 1 as an ensemble does not have a density function available in analytical form. We approximate its appearance in the logarithm term of the KL divergence by a normal density $f_\rho(s, \pi(s)) := \mathcal{N}(\mu_\pi(s), \sigma_\pi^2(s))$ where $\mu$ is calculated as in Step 3 but on the learned critic networks $X_i$ instead of their targets and $\sigma_\pi^2(s) = \frac{1}{K-1} \sum_{k=1}^{K} (X_k(s, \pi(s)) - \mu_\pi(s))^2$ on the current state $s$ instead of the next state $s'$. Although $\mathrm{KL}\left(f_\rho \parallel f_{\rho_p}\right)$ has an analytical solution, we choose the following approximation to ensure that the computation stays closer to the true posterior:

$$\mathrm{KL}\left(\rho(s, \pi(s)) \parallel \rho_0(s, \pi(s))\right) \approx \frac{1}{2K} \sum_{k=1}^{K} \left( \frac{(r + \gamma \bar{\mu}_\pi(s') - X_k)^2}{\gamma^2 \sigma_0^2} - \log \sigma_\pi^2(s) \right) + \text{const.}$$

See Appendix A for the full details of the derivation.

**5)** Since $\log(x) < x, \forall x \in \mathbb{R}^+$, taking the logarithm of the expected variance term in Eq. 2 makes the right-hand side of the inequality larger. Hence, it keeps the bound still valid. We use this version to ensure that the impact of increased ensemble element variances on the loss diminishes gradually. We observed this approach to improve performance, although the former option also learns.

**6)** We apply bootstrapping to detach the correlation between the ensemble elements, which has significant practical benefits demonstrated in prior work on deep exploration (Osband et al., 2016a; 2018). We pass each minibatch through a random binary mask which effectively filters some portion of the data for each ensemble element.

Putting all the pieces together, we arrive at the critic training objective

$$L(\{X_k : k \in [K]\}) := \underbrace{\frac{1}{nK} \sum_{i=1}^{n} \sum_{k=1}^{K} b_{ik} \left( r_i + \gamma \bar{X}_k(s_i', \pi(s_i')) - X_k(s_i, \pi(s_i)) \right)^2}_{\text{Diversity}}$$

$$+ \underbrace{\frac{1}{nK} \sum_{i=1}^{n} \sum_{k=1}^{K} \frac{b_{ik} \left( r_i + \gamma \bar{\mu}_\pi(s_i') - X_k(s_i, \pi(s_i)) \right)^2}{2\gamma^2 \sigma_0^2}}_{\text{Coherence}} - \underbrace{\frac{\gamma^2 + \frac{1}{2}}{n} \sum_{i=1}^{n} \log \sigma_\pi^2(s_i)}_{\text{Propagation}} \tag{3}$$

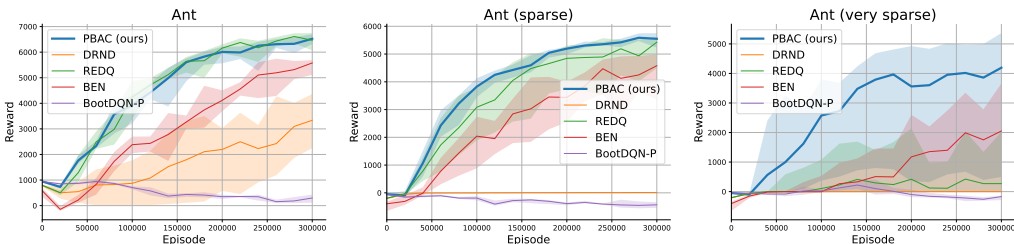

Figure 2: *Ant reward curves.* As we increase the sparsity of the reward, the models start to struggle to learn, with PBAC remaining most stable. See Figure 4 for the remaining reward curves.

for binary bootstrap masks $b_{ik} \sim \text{Bernoulli}(1-\kappa)$, $\forall [n] \times [K]$ for a bootstrapping rate $\kappa \in (0, 1)$.[3] The loss is computed by one forward pass through each ensemble member and their targets per data point with additional constant-time operations. Hence its computation time is similar to any other actor-critic method. The loss comprises three terms with interpretable roles. The first induces *diversity* into the ensemble by training each member on its individual targets. This way the model quantifies the uncertainty of its predictions as observations assigned to similar action values can only be those all ensemble elements are familiar with. The second term ensures *coherence* among ensemble elements as it trains them with the same target. The third term is a regularizer that repels ensemble elements away from each other in the absence of counter-evidence. This uncertainty *propagation* is maintained, as the model cannot find a solution to collapse the ensemble elements into a single solution. Propagation of uncertainties across time steps is known to have a critical role in deep exploration (Osband et al., 2018). As the variance increases, the PBAC loss converges to BootDQN-P (Osband et al., 2018). As it shrinks, it converges to the deterministic policy gradient.

Note that the PBAC critic training loss also nicely unifies the advantages of variational Bayesian inference (Blei et al., 2017) and Bayesian deep ensembles (Lakshminarayanan et al., 2017) into a single theoretically backed framework. Unlike variational Bayes, it works for density-free posteriors and unlike Bayesian deep ensembles, it permits a justified use of prior regularization via the Kullback-Leibler divergence term.

**Actor training and behavior policy.** We devise an actor network with a trunk $g(s)$ shared across the ensemble and individual heads $\pi_k(s) := (h_k \circ g)(s)$ for each ensemble element. Since the optimal decision rule under uncertainty is the Bayes predictor, we train the actor on the average value of each state with respect to the learned posterior $\arg\max_\pi \mathbb{E}_{s \sim P_\pi} \mathbb{E}_{X \sim \rho} X(s, \pi(s))$ and implement its empirical approximation as $\arg\max_{g, h_1, ..., h_K} \frac{1}{nK} \sum_{i=1}^{n} \sum_{k=1}^{K} X_k(s, \pi_k(s))$. We perform posterior sampling for exploration counting on its demonstrated theoretical benefits (Strens, 2000; Osband et al., 2013; Grande et al., 2014) and practical success in discrete control (Osband et al., 2016a; 2018; Fellows et al., 2021). That is, our behavior policy is randomly chosen among the available $K$ options and fixed for a number of time steps as practiced in prior work (Touati et al., 2020). In critic training, the Bellman targets are computed with the active behavior policy. See Appendix B.4 for details of how the hyperparameters of our model are chosen and the pseudocode in Appendix C for further algorithmic details.

## 4 EXPERIMENTS

With the experimental evaluation, we aim to validate our theoretical claims on improved performance for challenging deep exploration tasks with sparse rewards. Focusing on environments with nonlinear dynamics and continuous state and action spaces, we rely on a mixture of established sparse benchmarks from the DMC suite (Tassa et al., 2018) and introduce new sparse reward benchmarks based on various MuJoCo environments (Todorov et al., 2012). The aim is to evaluate whether PBAC improves performance in sparse reward setups while remaining competitive on their original standard dense reward counterparts. In Appendix B.1 we discuss prior work on sparse environments.

---

[3]We calculate $\bar{\mu}_\pi$ and $\sigma_\pi^2$ after applying the bootstrap masks.

Table 1: *Main results.* InterQuartile Mean (IQM) scores over ten seeds with [lower, upper] quantiles. A higher IQM indicates better performance. For each task, we make the IQM values bold if they fall within the range of the highest IQM score.

(a) IQM of the final episode reward

| ENVIRONMENT | BEN | BootDQN-P | DRND | REDQ | PBAC (ours) |
|---|---|---|---|---|---|
| ant | $5579_{[5114,5685]}$ | $298_{[114,430]}$ | $3337_{[2250,4363]}$ | $\mathbf{6464}_{[\mathbf{6069,6744}]}$ | $\mathbf{6524}_{[\mathbf{6316,6699}]}$ |
| hopper | $1051_{[567,1443]}$ | $\mathbf{1988}_{[\mathbf{1260,2494}]}$ | $\mathbf{1864}_{[\mathbf{1103,2914}]}$ | $\mathbf{1754}_{[\mathbf{1378,2663}]}$ | $\mathbf{1828}_{[\mathbf{1226,2918}]}$ |
| humanoid | $1706_{[1402,1909]}$ | $311_{[243,372]}$ | $5144_{[4743,5879]}$ | $\mathbf{5801}_{[\mathbf{5269,6226}]}$ | $819_{[598,1355]}$ |
| ballincup | $\mathbf{977}_{[\mathbf{971,980}]}$ | $973_{[965,978]}$ | $973_{[970,977]}$ | $\mathbf{980}_{[\mathbf{977,981}]}$ | $\mathbf{979}_{[\mathbf{975,983}]}$ |
| cartpole | $796_{[186,836]}$ | $601_{[379,770]}$ | $0.0_{[0.0,0.0]}$ | $0.0_{[0.0,0.0]}$ | $\mathbf{842}_{[\mathbf{838,845}]}$ |
| reacher | $\mathbf{-3.7}_{[\mathbf{-4.1,-3.5}]}$ | $-4.3_{[-4.6,-4.0]}$ | $-4.2_{[-4.3,-3.9]}$ | $\mathbf{-3.7}_{[\mathbf{-4.0,-3.5}]}$ | $\mathbf{-3.8}_{[\mathbf{-4.1,-3.7}]}$ |
| ant (sparse) | $4580_{[3369,4980]}$ | $-432_{[-542,-339]}$ | $10.0_{[8.3,12.7]}$ | $5427_{[4492,5617]}$ | $\mathbf{5549}_{[\mathbf{5229,5754}]}$ |
| ant (very sparse) | $\mathbf{2051}_{[\mathbf{-1,3661}]}$ | $-169_{[-291,-136]}$ | $-2.8_{[-2.8,-2.4]}$ | $269_{[-1,2019]}$ | $\mathbf{4194}_{[\mathbf{495,5365}]}$ |
| hopper (sparse) | $\mathbf{813}_{[\mathbf{689,928}]}$ | $\mathbf{1067}_{[\mathbf{659,1340}]}$ | $440_{[295,525]}$ | $\mathbf{879}_{[\mathbf{819,934}]}$ | $\mathbf{924}_{[\mathbf{772,1457}]}$ |
| hopper (very sparse) | $190_{[6,643]}$ | $\mathbf{388}_{[\mathbf{-0,1579}]}$ | $82_{[-0,347]}$ | $\mathbf{822}_{[\mathbf{774,918}]}$ | $\mathbf{852}_{[\mathbf{385,1636}]}$ |
| humanoid (sparse) | $6.0_{[5.9,6.1]}$ | $-3.3_{[-13.6,4.6]}$ | $5.9_{[5.8,5.9]}$ | $156_{[139,215]}$ | $\mathbf{1694}_{[\mathbf{1362,1896}]}$ |

(b) IQM of the aea under learning curve

| ENVIRONMENT | BEN | BootDQN-P | DRND | REDQ | PBAC (ours) |
|---|---|---|---|---|---|
| ant | $3103_{[2719,3387]}$ | $511_{[486,547]}$ | $1799_{[596,2613]}$ | $\mathbf{4700}_{[\mathbf{4422,5043}]}$ | $\mathbf{4635}_{[\mathbf{4353,4831}]}$ |
| hopper | $839_{[733,986]}$ | $974_{[949,1005]}$ | $952_{[736,1365]}$ | $\mathbf{1650}_{[\mathbf{1471,1817}]}$ | $\mathbf{1731}_{[\mathbf{1580,1900}]}$ |
| humanoid | $1227_{[1118,1295]}$ | $258_{[240,265]}$ | $\mathbf{2677}_{[\mathbf{2416,3026}]}$ | $\mathbf{2749}_{[\mathbf{2659,2929}]}$ | $656_{[532,785]}$ |
| ballincup | $\mathbf{954}_{[\mathbf{936,963}]}$ | $801_{[778,861]}$ | $863_{[773,951]}$ | $\mathbf{947}_{[\mathbf{874,965}]}$ | $923_{[835,967]}$ |
| cartpole | $423_{[179,513]}$ | $527_{[412,609]}$ | $0.0_{[0.0,0.0]}$ | $0.0_{[0.0,0.0]}$ | $\mathbf{650}_{[\mathbf{617,695}]}$ |
| reacher | $-4.2_{[-4.3,-4.1]}$ | $-4.6_{[-4.6,-4.5]}$ | $-4.4_{[-4.5,-4.3]}$ | $\mathbf{-4.1}_{[\mathbf{-4.1,-4.0}]}$ | $-4.3_{[-4.4,-4.2]}$ |
| ant (sparse) | $2705_{[1804,3126]}$ | $-307_{[-319,-288]}$ | $-0.9_{[-3.8,1.6]}$ | $3636_{[2430,4237]}$ | $\mathbf{3998}_{[\mathbf{3881,4179}]}$ |
| ant (very sparse) | $875_{[-16,1494]}$ | $-16_{[-76,53]}$ | $-7.6_{[-9.0,-6.0]}$ | $222_{[-7,1176]}$ | $\mathbf{2953}_{[\mathbf{397,3864}]}j$ |
| hopper (sparse) | $611_{[554,734]}$ | $275_{[170,394]}$ | $309_{[267,368]}$ | $634_{[552,781]}$ | $\mathbf{1163}_{[\mathbf{1068,1226}]}$ |
| hopper (very sparse) | $367_{[72,537]}$ | $62_{[-0,248]}$ | $55_{[-0,250]}$ | $509_{[421,569]}$ | $\mathbf{766}_{[\mathbf{651,1109}]}$ |
| humanoid (sparse) | $0.6_{[0.2,1.2]}$ | $-0.9_{[-3.5,1.9]}$ | $4.9_{[4.9,5.0]}$ | $99_{[84,120]}$ | $\mathbf{548}_{[\mathbf{480,705}]}$ |

To model reward sparsity in the MuJoCo environments, we explore the following two variations:

*(i)* A `sparse` version in which the health reward is always set to zero, therefore removing any incentive for the agent to maintain stability. This can cause the agent to prioritize speed (minimizing positional delay), potentially leading to inefficient movement patterns such as falling, rolling, or chaotic behavior. Without incentives for stability, the agent may fail to maintain balance or learn proper locomotion, resulting in suboptimal performance, harder training, and reduced overall effectiveness.

*(ii)* A `very-sparse` version in which the forward reward proportional to velocity is granted only if the agent reaches a distance threshold $c$. The degree of sparsity, hence the task difficulty, increases proportionally to $c$. This level of sparsity is applied on top of the zero health reward from the `sparse` modification. Although this incentivizes reaching the target, the lack of intermediate rewards for progress may lead the agent to struggle with exploration, resulting in slower learning and difficulty in discovering effective policies as the agent infrequently gains informative signals from the environment.

In these settings, the agent does not receive any positive reward and pays a penalty for using its actuators for failed explorations. Hence, they are well-suited for testing how directed the exploration scheme of a learning algorithm is. We use them to assess whether an exploration strategy can consistently promote effective exploration, when the agent encounters a lack of informative feedback during learning.

We build an ensemble of ten Q-functions and use a replay ratio of five to improve sample efficiency. This approach has been shown to achieve the same level of performance accuracy as state-of-the-art methods, effectively addressing sample efficiency, as demonstrated in Nauman et al. (2024). More details on the reward structure of each environment and the remaining design choices are provided in Appendix B. We provide a public implementation at `anonymous`.

**Baseline models.** We compare our method with other state-of-the-art exploration approaches on three standard MuJoCo tasks and their sparse versions along with three sparse DMC tasks.

REDQ (Chen et al., 2021) serves as a state-of-the-art representative of ensemble-based maximum entropy methods. We choose SAC-DRND (Yang et al., 2024) as the best representative of the RND family integrated to SAC, and BootDQN-P (Osband et al., 2018) as a close SOTA Bayesian model-free method to our model. Finally, BEN (Fellows et al., 2024) as the best representative approach that can learn the Bayes-optimal policies. More details about the selected baselines and design choices can be found in Appendix B.

**Main results.** We summarize our results in Table 1, reporting the interquartile mean (IQM) (Agarwal et al., 2021) together with the corresponding interquartile range over ten seeds. Reported are both the reward on the final episode, as well as the area under learning curve (AULC) for the whole training period, which quantifies the convergence speed. See Figure 2 for reward curves on the ant environment throughout increasing sparsity. We provide the corresponding curves for all environments in Figure 4. PBAC is competitive in most environmental settings and excels especially in the learning speed (subtable (b)) in sparse environments.

**Effects of hyperparameters.** The three main hyperparameters of PBAC are the bootstrap rate (BR) $\kappa$, the posterior sampling rate (PSR), and the prior variance (PV) $\sigma_0$ of $\rho_0$. An increase in the bootstrap rate enforces a higher diversity among the ensemble members. Decreasing the prior variance instead reduces diversity by pushing them towards a common mean. Lastly, the posterior sampling rate allows us to finetune the amount of exploration. While increasing the bootstrap rate and decreasing in the posterior sampling rate considerably slows down the learning process, PBAC remains robust to changes in the prior variance. We visualize their relative influence on the learning process using the cartpole environment conceptually in Figure 3. See Figure 7 and Figure 8 for more detailed figures on the cartpole and sparse ant environments.

**Exploration patterns.** Figure 1 shows how PBAC explores the first two dimensions of the state space (position and angle) in the cartpole environment, throughout its training process. We describe this visualization in greater detail and include the remaining methods as well as a second environment (sparse ant) in Appendix D.2.

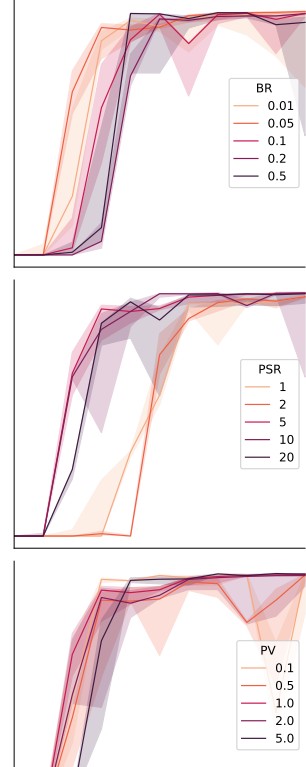

Figure 3: *Ablation.* Varying bootstrap rate (BR), posterior sampling rate (PSR) and prior variance (PV) on cartpole.

## 5 CONCLUSION

We introduced a method to do deep exploration in sparse reward environmental settings for the first time with a principled PAC-Bayesian approach. Comparing it to various state-of-the-art baselines, we demonstrated its superior performance on a wide range of continuous control benchmarks with various sparse reward patterns.

As the goal of our proposal is primarily to solve the task of deep exploration, PBAC's performance is sensitive to hyperparameters in the dense reward environments. While performing competitively in two of them, *ant* and *hopper*, it struggles, as do several of the baselines in the dense *humanoid* environment. This is due to the very high health reward of $r = 5$ an agent receives after every step, compared to $r = 1$ in the other two. It is not surprising that in this situation random exploration is sufficient and more robust. However, as soon as the reward becomes sparse, PBAC remains the only method to learn the task.

A theoretical limitation is that our current work does not provide any convergence guarantees on PBAC's behavior. This theoretical problem requires, and deserves, dedicated investigation. As PBAC builds on a sum of two Bellman backups and a regularizer, it will inherit similar properties to the convergence guarantees of stochastic iterative $Q$-learning variants. Generalization of these guarantees to continuous state spaces is known to be a nontrivial problem which we consciously keep outside our focus.

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

# APPENDIX

## A    PROOFS AND DERIVATIONS

For practical purposes, we assume in the proofs that all probability measures except $\rho$ have densities. The proofs can be straightforwardly extended by lifting this assumption.

**Lemma 1.** For any $\rho$ defined on $X$, the following identity holds

$$\widetilde{L}(\rho) = \mathbb{E}_{X \sim \rho} ||T_\pi X - X||_{P_\pi}^2 + \gamma^2 \mathbb{E}_{X \sim \rho}[\mathrm{Var}_{s \sim P_\pi}[X(s, \pi(s))]].$$

*Proof of Lemma 1.* For a fixed $X$ we have

$$\widetilde{L}(X) = \mathbb{E}_{s \sim P_\pi}\left[\mathbb{E}_{s' \sim P(\cdot|s,\pi(s))}\left[\left(\widetilde{T}_\pi X(s, \pi(s), s') - X(s, \pi(s))\right)^2 \Big| s\right]\right]$$

$$= \mathbb{E}_{s \sim P_\pi}\Big[\mathbb{E}_{s' \sim P(\cdot|s,\pi(s))}\big[\widetilde{T}_\pi X(s, \pi(s), s')^2$$
$$+ X(s, \pi(s))^2 - 2\widetilde{T}_\pi X(s, \pi(s), s')X(s, \pi(s))|s\big]\Big]$$

$$= \mathbb{E}_{s \sim P_\pi}\Big[\mathbb{E}_{s' \sim P(\cdot|s,\pi(s))}\left[\widetilde{T}_\pi X(s, \pi(s), s')^2|s\right]$$
$$+ X(s, \pi(s))^2 - 2\mathbb{E}_{s' \sim P(\cdot|s,\pi(s))}\left[\widetilde{T}_\pi X(s, \pi(s), s')|s\right]X(s, \pi(s))\Big]$$

$$= \mathbb{E}_{s \sim P_\pi}\Big[\mathbb{E}_{s' \sim P(\cdot|s,\pi(s))}\left[\widetilde{T}_\pi X(s, \pi(s), s')|s\right]^2$$
$$+ \mathrm{Var}_{s' \sim P(\cdot|s,\pi(s))}\left[\widetilde{T}_\pi X(s, \pi(s), s')|s\right] + X(s, \pi(s))^2$$
$$- 2\mathbb{E}_{s' \sim P(\cdot|s,\pi(s))}\left[\widetilde{T}_\pi X(s, \pi(s), s')|s\right]X(s, \pi(s))\Big]$$

$$= \mathbb{E}_{s \sim P_\pi}\bigg[\left(\mathbb{E}_{s' \sim P(\cdot|s,\pi(s))}\left[(\widetilde{T}_\pi X(s, \pi(s), s')|s\right] - X(s, \pi(s))\right)^2$$
$$+ \mathrm{Var}_{s' \sim P(\cdot|s,\pi(s))}\left[\widetilde{T}_\pi X(s, \pi(s), s')|s\right]\bigg]$$

$$= \mathbb{E}_{s \sim P_\pi}\bigg[\left(T_\pi X(s, \pi(s)) - X(s, \pi(s))\right)^2 + \mathrm{Var}_{s' \sim P(\cdot|s,\pi(s))}\left[\widetilde{T}_\pi X(s, \pi(s), s')|s\right]\bigg]$$

$$= ||T_\pi X - X||_{P_\pi}^2 + \mathbb{E}_{s \sim P_\pi}\left[\mathrm{Var}_{s' \sim P(\cdot|s,\pi(s))}\left[\widetilde{T}_\pi X(s, \pi(s), s')|s\right]\right]$$

$$= ||T_\pi X - X||_{P_\pi}^2 + \mathbb{E}_{s \sim P_\pi}\left[\mathrm{Var}_{s' \sim P(\cdot|s,\pi(s))}\left[r(s, \pi(s)) + \gamma X(s', \pi(s'))|s\right]\right]$$

$$= ||T_\pi X - X||_{P_\pi}^2 + \gamma^2 \mathbb{E}_{s \sim P_\pi}\left[\mathrm{Var}_{s' \sim P(\cdot|s,\pi(s))}\left[X(s', \pi(s'))|s\right]\right].$$

Since the Markov process is stationary we get

$$\mathbb{E}_{s \sim P_\pi}\left[\mathbb{E}_{s' \sim P(\cdot|s,\pi(s))}\left[X(s', \pi(s'))|s\right]\right] = \mathbb{E}_{s \sim P_\pi}\left[X(s, \pi(s))\right]$$

as well as

$$\mathbb{E}_{s \sim P_\pi}\left[\mathbb{E}_{s' \sim P(\cdot|s,\pi(s))}\left[X(s', \pi(s'))^2|s\right]\right] = \mathbb{E}_{s \sim P_\pi}\left[X(s, \pi(s))^2\right].$$

Hence, the variance term can be expressed as

$$\mathbb{E}_{s \sim P_\pi}\left[\mathrm{Var}_{s' \sim P(\cdot|s,\pi(s))}\left[X(s', \pi(s'))|s\right]\right] = \mathbb{E}_{s \sim P_\pi}\left[X(s, \pi(s))^2\right] - \mathbb{E}_{s \sim P_\pi}\left[X(s, \pi(s))\right]^2$$
$$= \mathrm{Var}_{s \sim P_\pi}\left[X(s, \pi(s))\right].$$

Taking the expectation over $X$ with respect to $\rho$ concludes the proof. $\qquad\square$

**Lemma 2.** *For any $Q_1, Q_2 : \mathcal{S} \times \mathcal{A} \to \mathbb{R}$ and stationary state visitation distribution $P_\pi$, the following inequality holds:*

$$||T_\pi Q_1 - T_\pi Q_2||_{P_\pi} \leq \gamma ||Q_1 - Q_2||_{P_\pi}.$$

*That is, the Bellman operator $T_\pi$ is a $\gamma$-contraction with respect to the $P_\pi$-weighted $L_2$-norm $||\cdot||_{P^\pi}$*

*Proof of Lemma 2.*

$$\|T_\pi Q_1 - T_\pi Q_2\|_{P_\pi}^2 = \mathbb{E}_{s \sim P_\pi}\left[\left(T_\pi Q_1(s, \pi(s)) - T_\pi Q_2(s, \pi(s))\right)^2\right]$$

$$= \gamma^2 \mathbb{E}_{s \sim P_\pi}\left[\left(\mathbb{E}_{s' \sim P(\cdot|s, \pi(s))}\left[T_\pi Q_1(s', \pi(s')) - T_\pi Q_2(s', \pi(s'))|s\right]\right)^2\right]$$

$$\leq \gamma^2 \mathbb{E}_{s \sim P_\pi}\left[\mathbb{E}_{s' \sim P(\cdot|s, \pi(s))}\left[\left(T_\pi Q_1(s', \pi(s')) - T_\pi Q_2(s', \pi(s'))\right)^2\Big|s\right]\right] \quad \textit{(Jensen)}$$

$$= \gamma^2 \mathbb{E}_{s' \sim P_\pi}\left[\left(T_\pi Q_1(s', \pi(s')) - T_\pi Q_2(s', \pi(s'))\right)^2\right]. \quad \textit{(Stationarity)}$$

Taking the square-root of both sides yields the result. □

**Lemma 3.** *For any $\rho$ defined on $X$ the following inequality holds:*

$$\mathbb{E}_{X \sim \rho}\|X - Q_\pi\|_{P_\pi} \leq \frac{\mathbb{E}_{X \sim \rho}\|T_\pi X - X\|_{P_\pi}}{1 - \gamma}.$$

*Proof of Lemma 3.* For any fixed $X$ we have

$$\|X - Q_\pi\|_{P_\pi} = \|X - T_\pi X + T_\pi X - Q_\pi\|_{P_\pi}$$
$$= \|X - T_\pi X + T_\pi X - T_\pi Q_\pi\|_{P_\pi}$$
$$\leq \|X - T_\pi X\|_{P_\pi} + \|T_\pi X - T_\pi Q_\pi\|_{P_\pi} \quad \textit{(Triangle ineq.)}$$
$$\leq \|X - T_\pi X\|_{P_\pi} + \gamma\|X - Q_\pi\|_{P_\pi}. \quad \textit{(Lemma 2)}$$

Rearranging the terms and integrating over all $X$'s weighted by $\rho$ yields the result. □

**Theorem 2.** *For any posterior and prior measures $\rho, \rho_0 \in \mathcal{P}$, error tolerance $\delta \in (0, 1]$, and deterministic policy $\pi$, simultaneously, the following inequality holds with probability at least $1 - \delta$:*

$$\mathbb{E}_{X \sim \rho}\|Q_\pi - X\|_{P_\pi}^2 \tag{4}$$
$$\leq \frac{1}{(1 - \gamma)^2}\left(\widehat{L}_{\mathcal{D}}(\rho) + \frac{1}{n}\left(\text{KL}\left(\rho \parallel \rho_0\right) + \ln\frac{1}{\delta} + \frac{nR^2}{(1 - \gamma)^2}\right) - \gamma^2 \mathbb{E}_{X \sim \rho}\text{Var}_{s \sim P_\pi}\left[X(s, \pi(s))\right]\right).$$

*Proof of Theorem 2.* Choosing $d(a, b) = |a - b|$ and applying Theorem 1 we get with probability at least $1 - \delta$

$$\mathbb{E}_{h \sim \rho}[L(h)] \leq \mathbb{E}_{h \sim \rho}[\widehat{L}_{\mathcal{D}}(h)] + \frac{1}{n}\left(\text{KL}\left(\rho \parallel \rho_0\right) + \ln\left(\frac{1}{\delta}\mathbb{E}_{\mathcal{D} \sim P}\mathbb{E}_{h \sim \rho_0}e^{nd(\widehat{L}_{\mathcal{D}}(h), L(h))}\right)\right).$$

Since $(\widetilde{T}_\pi X(s, \pi(s), s') - X(s, \pi(s)))^2 \in [0, R^2/(1 - \gamma)^2]$ in the assumed discounted setup, we have

$$\ln\left(\frac{1}{\delta}\mathbb{E}_{\mathcal{D} \sim P}\mathbb{E}_{h \sim \rho_0}e^{nd(\widehat{L}_{\mathcal{D}}(h), L(h))}\right) \leq \ln\left(\frac{1}{\delta}\mathbb{E}_{\mathcal{D} \sim P}\mathbb{E}_{h \sim \rho_0}e^{n\frac{R^2}{(1-\gamma)^2}}\right) = \ln\frac{1}{\delta} + \frac{nR^2}{(1 - \gamma)^2}.$$

Plugging this result into the bound with $L(\rho) := \mathbb{E}_{X \sim \rho}[L(X)]$ and $\widehat{L}_{\mathcal{D}}(\rho) := \mathbb{E}_{X \sim \rho}[\widehat{L}_{\mathcal{D}}(X)]$ gives

$$L(\rho) \leq \widehat{L}_{\mathcal{D}}(\rho) + \frac{1}{n}\left(\text{KL}\left(\rho \parallel \rho_0\right) + \ln\frac{1}{\delta} + \frac{nR^2}{(1 - \gamma)^2}\right).$$

By Lemma 1 we get

$$\mathbb{E}_{X \sim \rho}\|T_\pi X - X\|_{P_\pi}^2$$
$$\leq \widehat{L}_{\mathcal{D}}(\rho) + \frac{1}{n}\left(\text{KL}\left(\rho \parallel \rho_0\right) + \ln\frac{1}{\delta} + \frac{nR^2}{(1 - \gamma)^2}\right) - \gamma^2 \mathbb{E}_{X \sim \rho}\text{Var}_{s \sim P_\pi}\left[X(s, \pi(s))\right].$$

Applying Lemma 3 on the left-hand side yields the intended result. □

## A.1 Derivation of the approximation to the KL divergence term

$$\mathrm{KL}\big(\rho(s,\pi(s))||\rho_0(s,\pi(s))\big) \approx \mathbb{E}_{X\sim\rho}[\log f_\rho(X|s,a) - \log f_{\rho_0}(X|s,a)]$$

$$\approx \frac{1}{K}\sum_{k=1}^{K} \log f_\rho(X_k|s,\pi_k(s)) - \log f_{\rho_0}(X_k|s,\pi_k(s))$$

$$= \frac{1}{2}\frac{1}{K}\sum_{k=1}^{K}\left(-\log\sigma_\pi^2(s) - \frac{(r+\gamma\mu_\pi(s)-X_k)^2}{\sigma_\pi^2(s)} + \log(\gamma^2\sigma_0^2) + \frac{(r+\gamma\bar\mu_\pi(s')-X_k)^2}{\gamma^2\sigma_0^2}\right)$$

$$= \frac{1}{2}\Bigg(-\frac{1}{K}\sum_{k=1}^{K}\log\sigma_\pi^2(s) - \frac{1}{\sigma_\pi^2(s)}\underbrace{\frac{1}{K}\sum_{k=1}^{K}(r+\gamma\mu_\pi(s)-X_k)^2}_{=\frac{K-1}{K}\sigma_\pi^2(s)}$$

$$+ \log(\gamma^2\sigma_0^2) + \frac{(r+\gamma\bar\mu_\pi(s')-X_k)^2}{\gamma^2\sigma_0^2}\Bigg)$$

$$= \frac{1}{2K}\sum_{k=1}^{K}\left(\frac{(r+\gamma\bar\mu_\pi(s')-X_k)^2}{\gamma^2\sigma_0^2} - \log\sigma_\pi^2(s)\right) + \mathrm{const.}$$

## B Experimental details

### B.1 Prior work on reward sparsity in continuous control

How to structure a sparse reward environment is a matter of debate. We identify two broad groups of environments in the current state-of-the-art literature, without claiming this to be an exhaustive survey. These consist either of native, i.e., inherent sparsity in the original environment or are custom modifications of dense reward environments made by the respective authors. We follow the naming convention for each of the environments as used in the Gymnasium (Towers et al., 2024) and DMControl (Tassa et al., 2018) libraries unless otherwise noted.

**i) Binary rewards based on proximity to a target state.** Houthooft et al. (2016) and Fellows et al. (2021) study Mountain Car continuous and sparse cartpole swing up. Fellows et al. (2021) reach a reward of around 500 in cartpole within more than 1.5 million environment interactions, far behind the levels we report in our experiments. Curi et al. (2020) study reacher, pusher, a custom sparsified version of reacher with a reward squashed around the goal state and action penalty with an increased share. Houthooft et al. (2016), Mazoure et al. (2019), and Amin et al. (2021) sparsify various MuJoCo locomotors by granting binary reward based on whether the locomotor reaches a target $x$-coordinate. This reward design has limitations: i) Since there is no action penalty, the locomotor does not need to aim at its actuators in the most efficient way, ii) Since there is no forward reward proportional to velocity, the locomotor's performance after reaching a target location becomes irrelevant. For instance, a humanoid can also fall down or stand still after reaching the target.

**ii) Increased action penalties.** Curi et al. (2020) study cartpole and MujoCo HalfCheetah with increased action penalties. Luis et al. (2023b) use pendulum swingup with nonzero reward only in the close neighborhood of the target angle. They also apply an action cost and perturb the pendulum angle with Gaussian white noise. They also report results on PyBullet Gym locomotors HalfCheetah, Walker2D, and Ant with dense rewards. Their follow-up work (Luis et al., 2023a) addresses a larger set of DMC environments where locomotors are customized to receive action penalties. The limitation of this reward design is that increasing the action penalty is not sufficient to sparsify the reward as the agent can still observe relative changes in the forward reward from its contributions to the total reward. In some MuJoCo variants, the agent also receives rewards on the health status, which is a signal about intermediate success against the sparse-reward learning goal. Furthermore, action penalties are typically exogenous factors determined by energy consumption in the real world, making the challenge artificial.

In Section 4, we report results on the most difficult subset of some natively sparse environments and devise new locomotion setups that overcome the aforementioned limitations. We discuss them in the next subsection.

## B.2 EXPERIMENT PIPELINE DESIGN

The whole experiment pipeline is implemented in PyTorch (Paszke et al., 2019, version 2.4.1). The experiments are conducted on eleven continuous control environments from two physics engines. MuJoCo (Todorov et al., 2012; Brockman, 2016), and DeepMind Control (DMControl) Suite (Tassa et al., 2018). All MuJoCo environments used are from version 4 of the MuJoCo suite (V4), and sparse DMControl environments.

### B.2.1 MUJOCO

From the two locomotors that can stand without control, we choose *ant* as it is defined on the $3D$ space compared to the $2D$ defined *halfcheetah*. From the two locomotors that have to learn to keep their balance, we choose *hopper* instead of *walker2d* as hopping is a more dexterous locomotion task than walking. We also choose *humanoid* as an environment where a $3D$ agent with a very large state and action space has to learn to keep its balance. The reward functions of the MuJoCo locomotion tasks have the following generic form:

$$r := \underbrace{\frac{dx_t}{dt}}_{\text{Forward Reward}} - \underbrace{w_a||a_t||^2}_{\text{Action cost}} + \underbrace{H}_{\text{Health reward}} .$$

*Forward reward* comes from the motion speed of the agent towards its target direction and drives the agent to move efficiently toward the goal. *Health reward* is an intermediate incentive an agent receives to maintain its balance. The *action cost* ensure that the agent solves the task using minimum energy. We implement sparsity to the locomotion environments in the following two ways using the following reward function template:

$$r_{\text{sparse}} := \underbrace{\frac{dx_t}{dt}\mathbf{1}_{x_t > c}}_{\text{Forward Reward}} - \underbrace{w_a||a_t||^2}_{\text{Action cost}} .$$

This function delays forward rewards until the center of mass of the locomotor $x_t$ reaches a chosen target position $c$, which we call as the *positional delay*. This reward also removes the healthy reward to ensure that the agent does not get any incentive by solving an intermediate task. It is possible to increase the sparsity of a task by increasing the positional delay. Detailed information on the sparsity levels we used in our experiments are listed in Table 2. For each environment, we chose the largest positional delay, i.e. maximum sparsity, where at least one model can successfully solve the task. Beyond this threshold, all models fail to collect positive rewards within 300000 environment interactions. The structure of these experiments follows the same structure as what is known as the $n$-chains thought experiment, which is studied extensively in theoretical work. The essential property is that there is a long period of small reward on which an agent can overfit. See, e.g., the discussion by Strens (2000) or Osband et al. (2018) for further details. Information on the state and action space dimentionalities for each of the three environments is available in Table 3.

### B.2.2 DEEPMIND CONTROL

From DMControl, we choose *ballincup*, *cartpole*, and *reacher*, as they have sparse binary reward functions given based on task completion. In *ballincup*, the task is defined as whether the relative position of the ball to the cup centroid is below a distance threshold. In the *cartpole*, it is whether cart position and pole angle are in respective ranges $(-0.25, 0.25)$ and $(0.995, 1)$. Finally, in the *reacher*, it is the the distance between the arm and the location of a randomly placed target coordinate. Information on the state and action space dimentionalities for each of the threee environments is available in Table 3. We did not consider the DMC locomotors as they use the same physics engine as MuJoCo and their reward structure is less challenging due to the absence of the action penalty and the diminishing returns given to increased velocities.

Table 2: *MuJoCo environment reward hyperparameters.* See the description in the text for an explanation on each of the parameters.

| TASK | Positional delay $c$ | Action cost weight $w_a$ | Health reward $H$ |
|---|---|---|---|
| ant | 0 | 5e-1 | 1 |
| ant (sparse) | 0 | 5e-1 | 0 |
| ant (very sparse) | 2 | 5e-1 | 0 |
| hopper | 0 | 1e-3 | 1 |
| hopper (sparse) | 0 | 1e-3 | 0 |
| hopper (very sparse) | 1 | 1e-3 | 0 |
| humanoid | 0 | 1e-1 | 5 |
| humanoid (sparse) | 0 | 1e-1 | 0 |

Table 3: *State and action space dimensionalities for MuJoCo (MJC) and DMControl (DMC).*

| | TASK | $|\mathcal{S}|$ | $|\mathcal{A}|$ |
|---|---|---|---|
| MJC | ant | 27 | 8 |
| | hopper | 11 | 3 |
| | humanoid | 376 | 17 |
| DMC | ballincup | 8 | 2 |
| | cartpole | 5 | 1 |
| | reacher | 11 | 2 |

## B.3 EVALUATION METHODOLOGY

**Performance metrics.** We calculate the *Interquartile Mean (IQM)* of the final episode reward and of the *area under learning curve (AULC)* as our performance scores where the former indicates how well the task has been solved and the latter is a measure of learning speed. The AULC is calculated using evaluation episodes after every 20,000 steps. We calculate these rewards over ten repetitions on different seeds, where each of the methods gets the same seeds. All methods, including our approach and the baselines, utilize the same warmup phase of 10,000 steps to populate the replay buffer before initiating the learning process.

## B.4 HYPERPARAMETERS AND ARCHITECTURES

**PBAC specific hyperparameters.** PBAC has three hyperparameters: bootstrap rate, a posterior sampling rate, and prior variance. We observe PBAC to work robustly on reasonably chosen defaults. See Appendix D.3 for an ablation on a range of these for the cartpole and sparse ant environments. We list the hyperparameters we used for each environment in Table 4.

**Shared hyperparameters and design choices.** We use a *layer normalization* (Ball et al., 2023) after each layer to regularize the network, and a *concatenated ReLU (CReLU)* activation function (Shang et al., 2016) instead of the standard ReLU activation which enhances the model by incorporating both the positive and negative parts of the input and concatenating the results. This activation leads to potentially better feature representations and the ability to learn more complex patterns. Moreover, we rely on a high *replay ratio (RR)* and a small *replay buffer* size, which reduce the agent's dependence on long-term memory and encourage it to explore different strategies. These are employed to improve the plasticity of the learning process. Recently, Nauman et al. (2024) showed that a combination of these design choices can overall greatly improve the agent learning ability. Additionally, we opted to use the Huber loss function for all baseline models after observing in our preliminary trials that it consistently provided performance advantages across different baselines. All design choices found advantageous for our model and not harmful to other have also been applied to the baselines. Table 5 provides details on the hyper-parameters and network configurations used in our experiments.

Table 4: *Hyperparameters specific to PBAC.* The chosen bootstrap rate (BR), posterior sampling rate (PSR), and prior variance (PV) for each of the eleven environments. Changes from the defaults are marked in bold.

| TASK | BR | PSR | PV |
|---|---|---|---|
| ant | 0.05 | **1** | 1.0 |
| hopper | **0.01** | **1** | **10.0** |
| humanoid | 0.05 | 5 | **2.0** |
| ballincup | 0.05 | 5 | 1.0 |
| cartpole | 0.05 | 5 | 1.0 |
| reacher | 0.05 | 5 | 1.0 |
| ant (sparse) | 0.05 | 5 | 1.0 |
| ant (very sparse) | 0.05 | 5 | 1.0 |
| hopper (sparse) | **0.1** | 5 | 1 |
| hopper (very sparse) | **0.1** | 5 | **0.1** |
| humanoid (sparse) | 0.05 | 5 | **2.0** |

Table 5: *Shared hyper-parameters.* Hyperparameters used by all methods.

| Hyper-parameter | Value |
|---|---|
| Evaluation episodes | 10 |
| Evaluation frequency | Maximum timesteps / 100 |
| Discount factor $(\gamma)$ | 0.99 |
| $n$-step returns | 1 step |
| Replay ratio | 5 |
| number-of-critic-networks | 10 |
| Replay buffer size | 100,000 |
| Maximum timesteps* | 300,000 |
| Number of hidden layers for all networks | 2 |
| Number of hidden units per layer | 256 |
| Nonlinearity | CReLU |
| Mini-batch size $(n)$ | 256 |
| Network regularization method | Layer Normalization (LN) (Ball et al., 2023) |
| Actor/critic optimizer | Adam (Kingma & Ba, 2015) |
| Optimizer learning rates $(\eta_\phi, \eta_\theta)$ | 3e-4 |
| Polyak averaging parameter $(\tau)$ | 5e-3 |

 * Ballincup, reacher, and cartpole use a reduced number of maximum steps. The former two use 100.000 and the latter 200.000.

Table 6: *Actor and critic architectures.* Here, $d_s$ and $d_a$ are the dimensionalities of the state and action spaces.

| Actor network | Critic network |
|---|---|
| Linear($d_s$, 256) | Linear($d_s + d_a$, 256) |
| Layer-Norm | Layer-Norm |
| CReLU() | CReLU() |
| Linear(256, 256) | Linear(256, 256) |
| Layer-Norm | Layer-Norm |
| CReLU() | CReLU() |
| Linear(256, $d_a$) | Linear(256,1) |

**Actor and critic networks.** Our implementation of PBAC along with proposed baselines share the architectural designs provided in Table 6 for each critic network in the ensemble and actor network. The quantities $d_s$ and $d_a$ denote the dimension of the state space and the action space, respectively. The output of the actor network is passed through a $\tanh(\cdot)$ function for deterministic actor networks used in PBAC, BEN, and BootDQN-P. We implemented the probabilistic actors of REDQ and DRNB as a *squashed Gaussian head* uses the first $d_a$ dimensions of its input as the mean and the second $d_a$ as the variance of a normal distribution.

### B.4.1 BASELINES

We compare PBAC against a range of state-of-the-art general purpose actor-critic methods that are all empowered by ensembles. All design choices mentioned above found advantageous for our model and not harmful to other have also been applied to the baselines. Below we also explain further changes compared to the original works that we found to be benficial in preliminary experiments.

**BEN.** Bayesian Exploration Networks (BEN), introduced by Fellows et al. (2024) serve as the best representative that can learn a Bayesian optimal policy and handle the exploration vs. exploitation tradeoff. We modify BEN by relying on Bayesian deep ensembles (Lakshminarayanan et al., 2017) instead of the normalizing flow-based approach used in the original work. An ensemble of $K - 1$ heteroscedastic critics learns a heteroscedastic univariate normal distribution over the Bellman target, while the $K$th critic, regularized by the ensemble guides the actor network.

**BootDQN-P.** Bootstrapped DQN with randomized prior functions (Osband et al., 2018), a Bayesian model-free approach, serve as a close relative to our method. Throughout all environments we share most parameters with PBAC. Changes for specific parameters are discussed in Table 7, and rely on a Thompson sampling actor. Its randomized priors allow the model to explore even in the presence of sparse reward. A prior scaling (PS) parameter regulates their influence.

**DRND.** Distributional randomized network distillation (DRND) (Yang et al., 2024), model the distribution of prediction errors from a random network. This distributional information is used as signal to guide exploration. As in the original work, we integrate it into a soft actor-critic (SAC) (Haarnoja et al., 2018) framework. This random predictor network is trained via the same objective and uses the same architecture as proposed by Yang et al. (2024). Actor and critic networks follow the architectural choices described above. Additionally, we optimize the $\alpha$ scaling parameter in the SAC as is common practice.

**REDQ.** Randomized ensemble double Q-learning (REDQ) (Chen et al., 2021) is the most competitive variant of the ensemble version of SAC as it incorporates both double Q-learning and randomization to address value estimation and exploration. For this method all design choices are aligned with our method. REDQ relies only on the shared parameters.

Table 7: *Hyperparameters specific to BootDQN-P.* The chosen bootstrap rate (BR), posterior sampling rate (PSR), and prior scaling (PV) for each of the eleven environments. Changes from the defaults are marked in bold.

| TASK | BR | PSR | PS |
|------|-----|-----|-----|
| ant | 0.05 | 5 | 5.0 |
| hopper | 0.05 | 5 | 5.0 |
| humanoid | **0.1** | **1** | 5.0 |
| ballincup | 0.05 | 5 | 5.0 |
| cartpole | 0.05 | **10** | **1.0** |
| reacher | 0.05 | **1** | **1.0** |
| ant (sparse) | **0.1** | **1** | **1.0** |
| ant (very sparse) | 0.05 | 5 | 5.0 |
| hopper (sparse) | 0.05 | 5 | 5.0 |
| hopper (very sparse) | 0.05 | 5 | 5.0 |
| humanoid (sparse) | **0.1** | **1** | **9.0** |

## C  PSEUDOCODE

We provide pseudocode for our model in Algorithm 1.

## D  FURTHER RESULTS

### D.1  REWARD CURVES

See Figure 4 for the full reward curves of all environments corresponding to the results presented in Table 1 in the main text.

### D.2  STATE SPACE VISUALIZATIONS

Throughout the training, we record the currently visited state at regular intervals and plot them in five groups, for PBAC and each of the baselines. We visualize two environments: cartpole and sparse ant. In each case we record every 500th step over the whole training process and record the corresponding state. The whole set is split into five groups and each is plotted as its own scatter plot.

#### D.2.1  CARTPOLE

Of cartpole's five state dimensions, only the first two are interpretable, giving us the position of the cart and the cosine of the angle of the pole. We visualize them in Figure 5. As discussed above, the agent gets rewarded only if it manages to stay close to the zero with an upwards pole, i.e., an angle close to one.

PBAC is able to quickly explore the state space and then concentrate on visiting the states with hight reward (i.e., the top middle). Similar to it, BootDQN-P has no problem in exploring the state space, however it never manages to find the narrow target and thus never converges. BEN starts exploring a wide range of states, but ultimately gets stuck in this seed without being able to find the target. As shown in Figure 4 BEN's performance varies greatly depending on the random initial seed in this environment. As such, this is a random representative of a failure case, not of its general performance on cartpole. DRND and REDQ quickly get stuck as well within a small subset of the state space, essentially just exploring the position of the cart within ever being able to significantly change the angle of the pole.

#### D.2.2  SPARSE ANT

The ant environment has a 27 dimensional state space. In order to properly visualize what is happening we rely on TSNE (Van der Maaten & Hinton, 2008) and compute a two dimensional embedding, which we visualize in five subsets as for the cartpole environment. Note that due to the inherent

---

**Algorithm 1** PAC-Bayesian Actor Critic (PBAC)

---

1: **Input:** Polyak parameter $\tau \in (0,1)$, mini-batch size $n \in \mathbb{N}$, bootstrap rate $\kappa$, posterior sampling rate $\texttt{PSR}$, prior variance $\sigma_0^2$, number of ensemble elements $K$
2: **Initialize:** replay buffer $\mathcal{D} \leftarrow \varnothing$, critic parameters $\{\theta_k\}$ and targets $\bar{\theta}_k \leftarrow \theta_k$, actor network trunk $g$ and heads $h_1, \ldots h_K$.
3: $s \leftarrow \texttt{env.reset()}$ and $e \leftarrow 0$ (interaction counter)
4: **while** training **do**
5:     **if** $\mod(e, \texttt{PSR}) = 0$ **then**
6:       $j \sim \text{Uniform}(K)$
7:       $\pi \leftarrow \pi_{g \circ h_j}(s)$ (Update active critic)
8:     **end if**
9:     $a \leftarrow \pi_j(s)$ and $(r, s') \leftarrow \texttt{env.step}(a)$ and $e \leftarrow e+1$
10:     Store new observation: $\mathcal{D} \leftarrow \mathcal{D} \cup (s, a, r, s')$
11:     Sample minibatch: $B \sim \mathcal{D}$ with $|B| = n$
12:     Sample a bootstrap mask: $b_{ik} \sim \text{Bernoulli}(1 - \kappa), \quad \forall [n] \times [K]$
13:     Compute prior mean and posterior moments: $\forall (s_i, a_i, r_i, s_i') \in B$ do

$$\bar{\mu}_{\pi_{g \circ h_j}}(s_i') \leftarrow \frac{1}{K} \sum_{k=1}^{K} b_{ik}(\bar{X}_k(s_i', \pi_{g \circ h_j}(s_i')))$$

$$\mu_{\pi_{g \circ h_j}}(s_i) \leftarrow \frac{1}{K} \sum_{k=1}^{K} b_{ik}(X_k(s_i, \pi_{g \circ h_j}(s_i)))$$

$$\sigma^2_{\pi_{g \circ h_j}}(s_i) \leftarrow \frac{1}{K-1} \sum_{k=1}^{K} b_{ik}(X_k(s_i, \pi_{g \circ h_j}(s_i)) - \mu_{\pi_{g \circ h_j}}(s_i))^2$$

14:     Update critics $k \in [K]$:

$$\theta_k \leftarrow \arg\min_{\theta_k} \left\{ \frac{1}{nK} \sum_{i=1}^{n} \sum_{k=1}^{K} b_{ik} \Big( r_i + \gamma \bar{X}_k(s_i', \pi_{g \circ h_j}(s_i')) - X_k(s_i, \pi_{g \circ h_j}(s_i)) \Big)^2 \right.$$

$$\left. + \frac{1}{nK} \sum_{i=1}^{n} \sum_{k=1}^{K} \frac{b_{ik} \Big( r_i + \gamma \bar{\mu}_{\pi_{g \circ h_j}}(s_i') - X_k(s_i, \pi_{g \circ h_j}(s_i)) \Big)^2}{2\gamma^2 \sigma_0^2} - \frac{\gamma^2 + 1/2}{n} \sum_{i=1}^{n} \log \sigma^2_{\pi_{g \circ h_j}}(s_i) \right\}$$

15:     Update actor:

$$\phi \leftarrow \arg\max_{g, h_1, \ldots, h_K} \left[ \frac{1}{nK} \sum_{i=1}^{n} \sum_{k=1}^{K} X_k(s_i, \pi_{g \circ h_k}(s_i)) \right]$$

16:     Update critic targets: $\bar{\theta}_k \leftarrow \tau \theta_k + (1 - \tau)\bar{\theta}_k$ for $k \in [K]$
17:     **if** episode end **then** $s \leftarrow \texttt{env.reset()}$ **else** $s \leftarrow s'$
18: **end while**

---

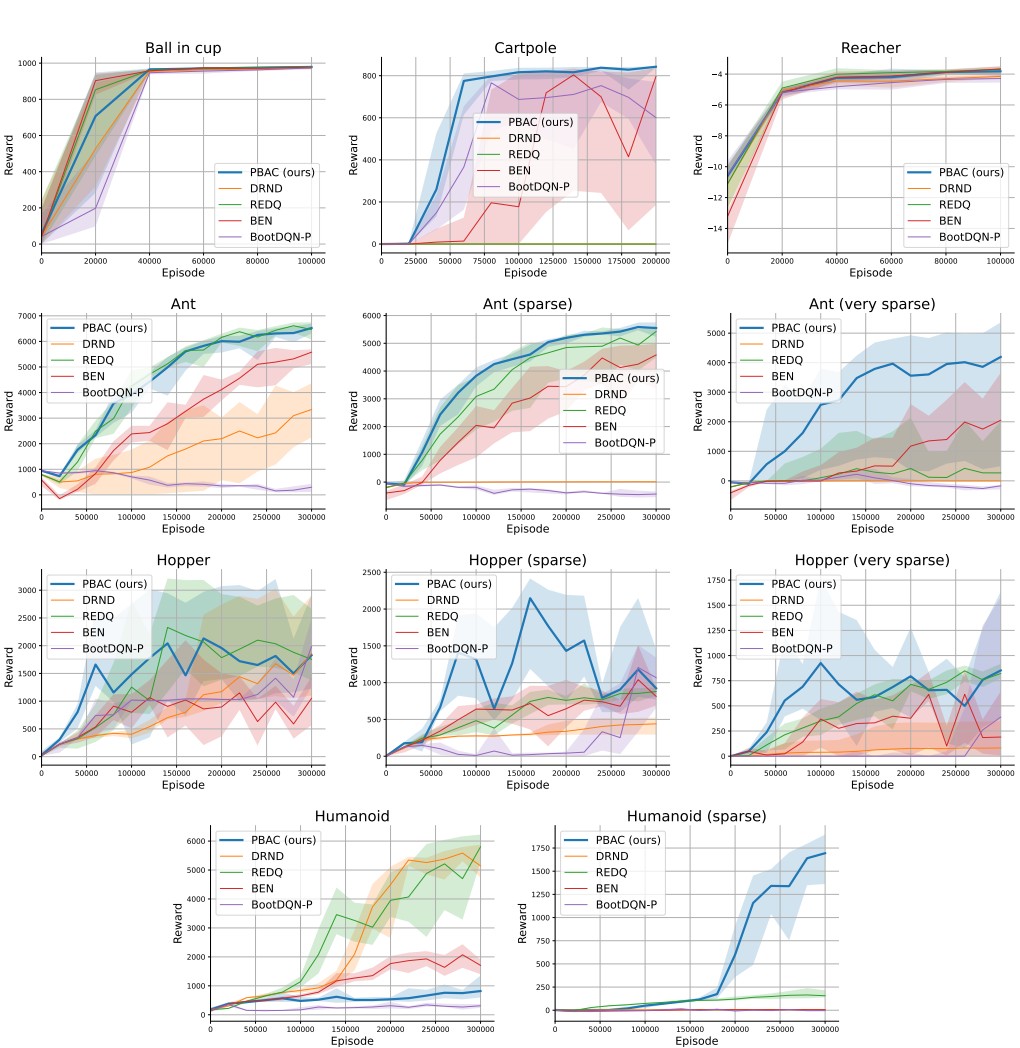

Figure 4: *Reward curves.* The reward curves for all environments throughout training corresponding to the results presented in Table 1. Visualized are the interquartile mean together with the interquartile range over ten seeds.

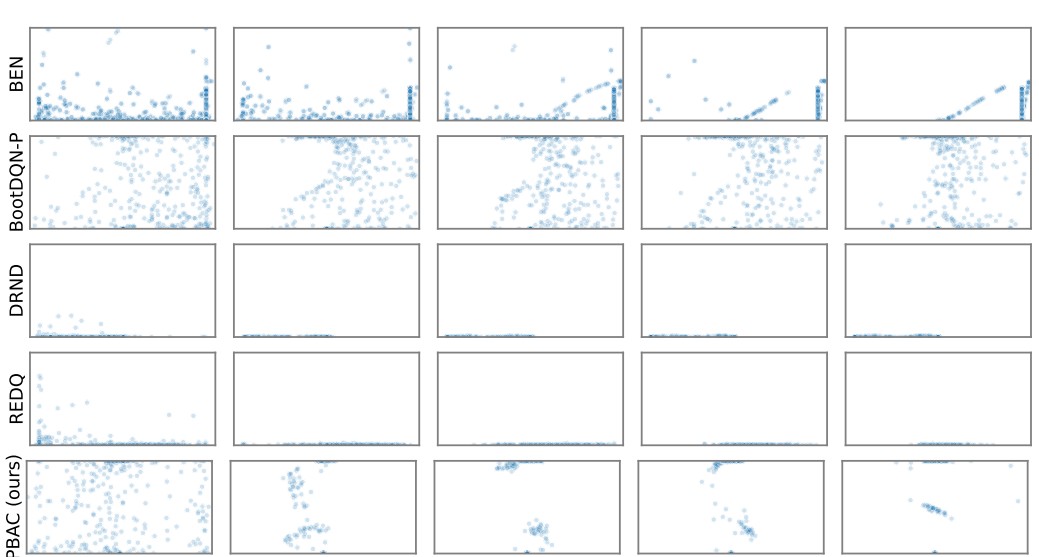

Figure 5: *State visitation frequency for cartpole.* The visualization shows seed 1.

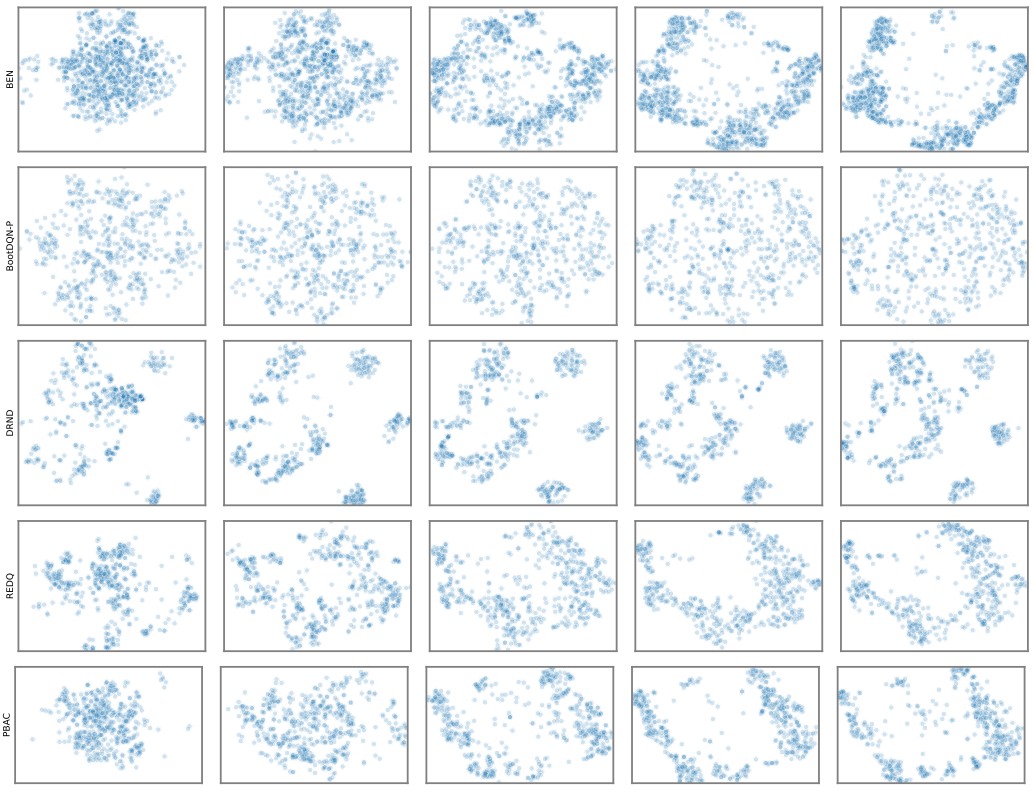

Figure 6: *State visitation frequency for sparse ant.* The visualiztion shows seed=1.

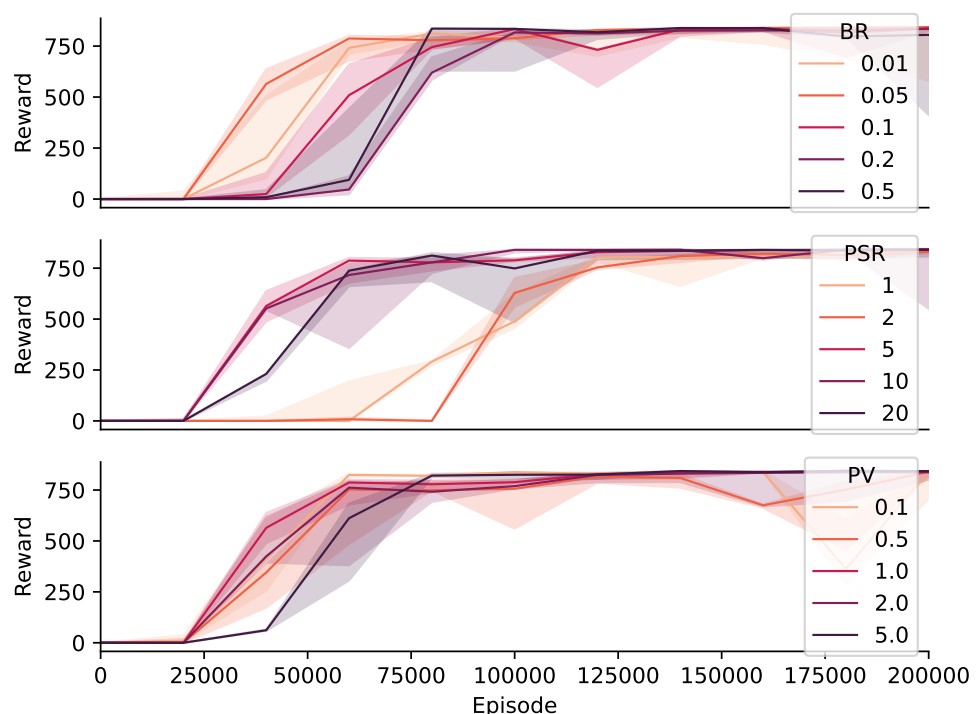

Figure 7: *Ablation results on cartpole.* We show varying bootstrap rates (BR), posterior sampling rates (PSR), and prior variances (PV). While PBAC is mostly robust in terms of varying prior variances, increases in the bootstrap rate and decreases in the posterior sampling rate delay the learning process. Visualized are the interquartile mean together with the interquartile range over three seeds.

stochastisity of TSNE, visual consistency only exists within the scatter plots of one environment (the whole sequence is mapped jointly), but not between the methods, as each learns its own transformation. As before we see that BootDQN-P seems to effortlessly explore a consistent area of the state space, however it never reaches an area of high reward.[4] DRND as well shows a similar pattern to its behavior in cartpole. It explores, but is again stuck into clusters that it can't escape from.[5] BEN, REDQ, and PBAC both show a similar pattern of exploration and subsequent exploitation.

### D.3   ABLATION

We evaluate the sensitivity of the training process of PBAC with respect to its three main hyperparameters, bootstrap rate, posterior sampling rate, and prior variance, on two environments. Depending on the environment, PBAC is sensitive to their choice (see Figure 7 on the cartpole environment), and shows clearly interpretable patterns, or it remains insensitive to their choice as in the sparse ant environment visualized in Figure 8.

---

[4]Due to the nature of TSNE the visualized area might be a large one of the original observation space, or a small one, i.e., the extend of the exploration can't be fully judged.

[5]The corresponding cluster in cartpole is around an angle of -1.

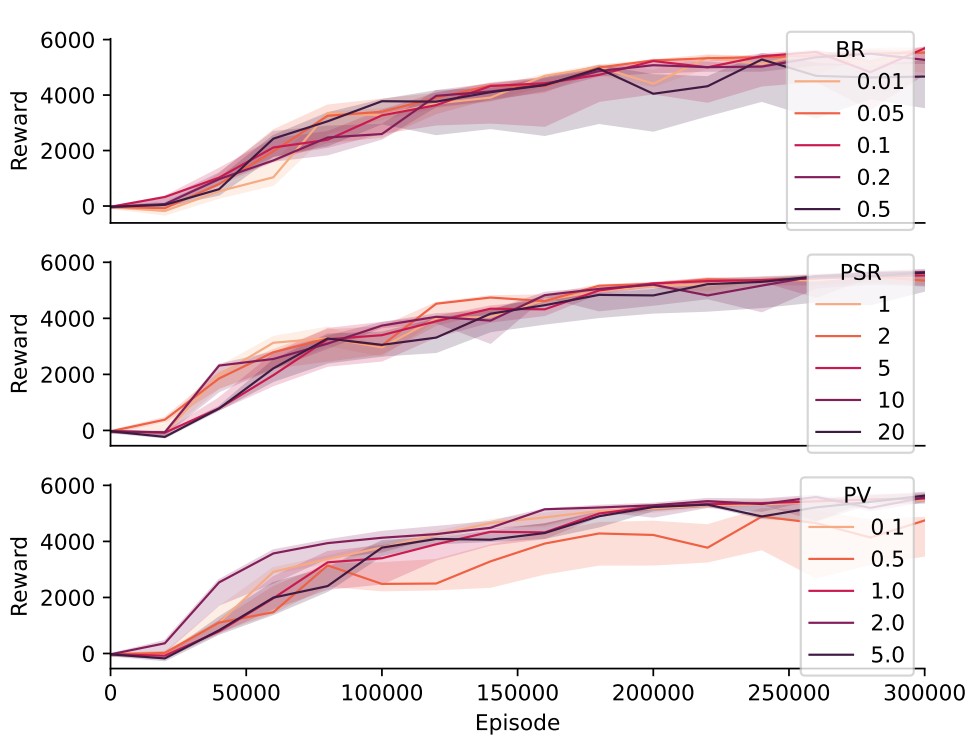

Figure 8: *Ablation results on sparse ant.* We show varying bootstrap rates (BR), posterior sampling rates (PSR), and prior variances (PV). Compared to the ablation on cartpole (see Figure 7), PBAC is robust against variations in all three hyperparameters in the sparse ant environment. Visualized are the interquartile mean together with the interquartile range over three seeds.

