# OpenReview forum: "Deep Exploration with PAC-Bayes"
_ICLR.cc/2025/Conference — Submitted to ICLR 2025_

### Official Review · Reviewer_dcYY · 2024-11-01

**Soundness:** 2
**Presentation:** 3
**Contribution:** 3
**Rating:** 5
**Confidence:** 3

**Summary:**

The paper presents a novel method for exploration in deep reinforcement learning. The method uses a PAC-Bayesian perspective to derive a novel critic update rule, which it then uses as part of an actor-critic setup. They actualize this using a bootstrapped ensemble of critics

**Strengths:**

Originality:
 * The method is the first to formulate the deep exploration problem from a PAC-Bayesian perspective and overall their method seems original.

Quality:
 * Overall the paper is well written. They achieve decent results, especially in their custom environments.

Significance:
 * Exploration continues to be a difficult problem in a variety of settings. I believe that the problem setting is well motivated and the paper has the potential to be significant from this perspective.
* The paper has some decent results

**Weaknesses:**

Some key issues:
 * The paper lacks any real analysis of the results. The main takeaway seems to be "our method is better" and that's it. Why does PBAC outperform baselines in the environments where it does? Why doesn't it see a similar improvement for the other benchmarks? Are there any other insights you can share
 * I find it concerning that the main place you see a win is in your own custom environments, because it's difficult to know if this win is because of your algorithm truly doing better or if the baselines could be tuned to achieve similar performance. I think it would be good to see some other common exploration benchmarks.

**Questions:**

Experiments questions:
 * Can you give more intuition for how to interpret the positional delay parameter $c$? You say in the paper that it is 2 and 1 for ant and humanoid respectively but what do these values mean? How many timesteps does it take for the agent to reach these distances? Can they be reached occasionally through random actions or do they absolutely require an exploration bonus?
 * Why no very sparse version for the humanoid environment?
 * How are hyperparameters tuned for comparison algorithms? Is there any chance that if you tuned their hyperparameters to the extent that you tuned yours then you would see similar performance? I see you tuned BootDQN-P but how about important exploration benchmarks like DRND?
 * Why does your method perform better on the sparse version of humanoid than the original?

Tiny notes / questions:
 * Figure 3 needs axis labels

---

> ### Author Response · Authors · 2024-11-14
> **Answer to dcYY**
>
> _Edit: Fixed a typo_
>
> Thank you for reviewing our submission and the provided feedback.
>
> - **Why does PBAC outperform baselines in the environments where it does? Why doesn't it see a similar improvement for the other benchmarks**
> The only two environments where PBAC does not perform as well as or better than the baselines are humanoid in both performance metrics and reacher in the area under learning curve. For humanoid, we hypothesize that this is due to the very high health reward the agent receives (5x as large as in the other dense environments). For reacher, while the difference is statistically significant, it is still very small in absolute reward.
>
> - **I think it would be good to see some other common exploration benchmarks.**
> Based upon the suggestion of reviewer v8fi, we are currently running experiments on various Meta-World environments and will update the general answer with those results once they are ready. Would you require further benchmarks?
>
> - **Can you give more intuition for how to interpret the positional delay parameter? How many timesteps does it take for the agent to reach these distances?**
> The _forward reward_ in MuJoCo environments depends on an agent's movement as measured in changes of the $x$ coordinate (See the `gymnasium` documentation for specifics on each environment). The delay parameter $c$ enforces that the agent only receives this reward if it has moved a distance of 1 or 2 from its starting position. As the hopper agent can fall over more easily, and thus end its episode, than an ant agent, we chose the required distance for the latter to be higher. See also Section B.2.1 for further details. As the humanoid environment is even more complex, even removing the health reward causes the baselines to fail. Further adding a positional delay restriction on the forward reward makes learning practically impossible.
>
> - **How are hyperparameters tuned for comparison algorithms?**
> To stay close to each other, all methods share the same optimizer and architectural backbone. _BootDQN-P_ and _PBAC_ contain several extra hyperparameters, summarized in Tables 4 & 7. Note that most of these parameters are constant throughout the experiments, i.e., the methods tend to be robust to them. _REDQ_ does not contain any extra hyperparameters that require tuning on top of the shared ones. _BEN's_ hyperparameters are also all shared apart from a similar bootstrap rate parameter which we kept fixed throughout after preliminary results showed it to be robust against it. Finally, for _DRND_ the $\alpha$ scaling parameter from SAC is optimized via gradient descent during training. Its main distinction from the other baselines and our proposal is its predictive network. As a large search over potential architectures for each environment would have been too costly, we followed the choices of the original authors, who kept them constant on a wide array of experiments indicating their robustness.
>
> - **Why does your method perform better on the sparse version of humanoid than the original?**
> As discussed in the conclusion, the high health reward (five times higher than in the other two MuJoCo environments) means that our method with its focus on exploration is unlikely to be able to compete. However, as soon as this health reward is dropped, the very sparse reward signal requires a powerful exploration method.
>
> We hope that these answers clarify your concerns. Please let us know if there are further points that should be discussed.

---

> ### Author Response · Authors · 2024-11-22
>
> Dear reviewer,
> As the discussion period will soon end, we are contacting you to double-check whether our answers resolved the concerns and questions you raised during your rebuttal. Due to the cross-discipline nature of our submission, i.e., the combination of PAC-Bayesian with RL approaches, we want to ensure that any potential communication gap that seems to be reflected in your low confidence score can be overcome.
> Please let us know if we can provide further details and clarification.

---

> > ### Comment · Area_Chair_noaN · 2024-11-24
> > **Please respond to rebuttal ASAP**
> >
> > Dear reviewer,
> > The process only works if we engage in discussion. Can you please respond to the rebuttal provided by the authors ASAP?

---

> > > ### Comment · Reviewer_dcYY · 2024-11-24
> > >
> > > Hi, apologies for the slow response.
> > >
> > > Thanks for the response and the additional experiments. Comments:
> > >  - I feel there is still a lack of any analysis. Part of the goal of the paper is to convey intuition to readers about why your method performs better in certain settings than comparisons. You provided some intuition about why your method doesn't do well for the humanoid environment but I still don't see anywhere where you've made a specific case for your method beyond just saying it's better because it achieved higher reward. Same thing with the new MetaWorld experiments. What insights do you get from them beyond "ours is better"?
> > >  - I found the means you used for presenting MetaWorld results to be quite contrived and difficult to interpret. Is there a reason you can't just update the draft with learning curves or some other more standard method for presenting RL results?
> > >  - I've read over the comments from other reviewers and tend to agree with the consensus that the results do not show enough of an improvement to justify the added complexity, and that the method has key limitations in areas such as denser reward settings and higher dimensional settings.

---

> > > > ### Author Response · Authors · 2024-11-25
> > > >
> > > > Thank you for your response and additional remarks.
> > > >
> > > > - As discussed in the paper, our policy evaluation relies on a function-space-based posterior sampling scheme, which provides better-calibrated uncertainty predictions as shown in the Bayesian deep learning literature. These allow PBAC to explore the state space better than our baselines, especially as the rewards get sparser and deep exploration is needed. Please note, that REDQ is the only method that remains somewhat competitive and it does so primarily as measured by the final episode reward, which, as was also argued for by reviewer v8fi, is a suboptimal performance metric. As soon as we measure performance by taking the whole reward curve into account, summarized by the area under learning curve measure, it performs significantly worse ($p < 0.05$, see our latest answer to v8fi for details on the testing procedure) in five out of eight sparse environments for MuJoCo, and worse on the meta-world experiment for four out of five.
> > > > - We don't consider _"time until an agent has reliably solved the task"_ to be complex or difficult to interpret. Could you specify which part of that measure is difficult? We chose this measure to be able to provide varied result on all baselines for a reasonable number of seeds to remove as much seed variance as possible in the short amount of time available during the rebuttal phase. Running a full set 1e6 environment steps for all five experiments and five baselines for at least three seeds (and preferably more) was not feasible due to computational limitations. We will of course, as we did for MuJoCo, provide full reward curves over ten seeds and 1e6 environment steps in a camera-ready version.
> > > > - Can you provide further details on which parts of the method you consider especially complex to improve our presentation? Section 3.2 provides detailed explanations of each part of the model which have all been shown to be important in their respective research domains.
> > > > - Concerning the reduced performance in dense reward settings. These were not the focus of our current work which focuses on sparse rewards. Given that only three dense reward environments were evaluated with PBAC performing significantly worse only in one such a conclusion seems premature. A full evaluation of the model's performance to warrant such a statement would require an extensive additional study, and leave the question as to why one would want to use a method designed for deep exploration in a reward-dense environment, rather than pick an algorithm designed for such a setup.
> > > > - In the high-dimensional humanoid environment, with a 348-dimensional observation space, PBAC is the only method able to solve the task as rewards become sparse. Given the current data, we do not consider it a true statement that PBAC has a key limitation in higher dimensional settings.

---

### Official Review · Reviewer_v8fi · 2024-11-06

**Soundness:** 2
**Presentation:** 2
**Contribution:** 2
**Rating:** 5
**Confidence:** 2

**Summary:**

This paper proposes an approach for deep exploration in sparse reward environments by applying a PAC-Bayesian framework in RL. Empirical results are presented in sparse reward environments however this method does not work in dense reward environments and lacks theoretical convergence guarantees.


Modeling the Posterior Distribution consists of the following pieces: The posterior distribution is represented as an ensemble of $K$ critic networks, each weighted equally. Distributions $\rho$ and $\rho_0$ are modeled directly in function space rather than weight space.
A data-informed prior $\rho_0$ is constructed based on the critic targets, providing the most recent information about action values. This prior is modeled as a normal distribution estimated from the critic targets. The KL divergence between $\rho$ and $\rho_0$
is then calculated using probability density functions evaluated at the outputs of the critic ensemble.

**Strengths:**

- The PAC-Bayesian framework provides a flexible way to assess generalization, allowing any posterior distribution to be optimized for data, making it highly adaptable. It's a nice area of research.
- The paper covers aspects of PAC-Bayesian RL, ensuring the bound’s validity and robustness in training.
- Applying the PAC-Bayesian framework for exploration in RL is novel and challenging. So I appreciate the effort.
- Empirical across various continuous control tasks with different reward structures.

**Weaknesses:**

- PBAC’s structured exploration approach is less effective in dense reward environments, particularly when high rewards are frequent. In these settings, random exploration often performs as well or better.
- The method lacks convergence guarantees,  limiting confidence in its stability and long-term performance across diverse tasks.
- Looking at the results, particularly the reward plots and Figure 4, it seems that the complexity of this method outweighs its advantages. The performance improvements shown aren’t particularly impressive, and given the considerable complexity involved.

**Questions:**

- There are several established sparse reward testbeds in robotics. I’m curious why the authors opted to modify Gym environments for sparse rewards. Sparse reward testbeds, such as Meta-World and D4RL, naturally simulate environments with sparse feedback. These frameworks offer realistic benchmarks that directly address sparse rewards, enabling comparison with other established state-of-the-art methods on these platforms without the need for manual modifications.

- Choice of IQM over Reward Plots: Could you elaborate on the decision to present performance results in Table 1 using IQM scores rather than traditional reward plots? I’m curious if this choice was intended to mitigate variability across seeds or to provide a more concise comparison across tasks. Would reward plots offer additional insights, or do IQM tables capture the main performance differences effectively? I understand that reward plots can sometimes be misleading due to high variability across seeds, especially in sparse reward environments where performance can vary significantly.


**Miscellaneous Comments**
- Table 1: In the table description, "IQM of the area under learning curve" should be corrected to "area under the learning curve."

---

> ### Author Response · Authors · 2024-11-14
> **ANswer to v8fi**
>
> Thank you for the detailed review of our submission.
>
> - **In dense reward environments random exploration performs as well or better.**
> That is to be expected. If the reward structure is dense, a method build for deep exploration looses its distinguishing feature.
> This is indeed the main motivation for performing deep exploration. See, e.g., the discussion in Osband et al. (2016a).
>
> - **The method lacks convergence guarantees, limiting confidence in its stability and long-term performance across diverse tasks.**
>     We acknowledge that we do not provide a theoretical convergence guarantee. Note, that neither does $Q$-learning for continuous state-action spaces, and consequently any deep actor-critic method. All guarantees assume finite state space size. See for instance Theorem 1 of the famous TD3 method proposed by Fujimoto et al. (2018) for a proof of convergence assuming a finite Markov decision process.  Once these assumptions are made, it is trivial to show that our method has identical convergence profile to any other actor-critic approach.. Currently only empirical results are available for its long-term performance and stability. We are currently increasing the diversity of our experiments by running further experiments on the Meta-World testbed as you suggested.
> - **Looking at the results, particularly the reward plots and Figure 4, it seems that the complexity of this method outweighs its advantages. The performance improvements shown aren’t particularly impressive, and given the considerable complexity involved.**
> We provide one-sided paired t-tests in the general answer above, which demonstrates significantly ($p<0.05$) improved performance for most experiments. Could you provide further details on where you see a _considerable complexity_? The final algorithm (Alg 1 on p23) can be implemented in a straight-forward manner that isn't more difficult than our reported baselines.
> - **I’m curious why the authors opted to modify Gym environments for sparse rewards. Sparse reward testbeds, such as Meta-World and D4RL, naturally simulate environments with sparse feedback.**
> Thank you for the suggestion. We are now running additional experiments on Meta-World and will provide them in a general answer once they are ready. Our decision to report custom environments was due to our aim to provide a set of experiments of increasing sparsity which a clearly defined reward structure (see Sec B.2.1).
>
> - **Could you elaborate on the decision to present performance results in Table 1 using IQM scores rather than traditional reward plots?**
> We do report the reward plots in Figure 2 and 4, which you also refer to in this review. Table 1 gives an additional concise scalar summary of these plots. Table 1(a) summarizes the results at the end of training, i.e., the end of each reward plot, while Table 1(b) summarizes the overall learning trajectory by computing its area under curve. We consider both, tabular and graphical presentation as important. The decision to include only a subset of the reward plots in the main text was only due to space restrictions. Could you clarify the question?
>
> We hope that this clarifies the concerns raised in this review. If there are further questions, please let us know.
>
>
> _____
> Fujimoto et al. 2018., _Addressing function approximation error in actor-critic methods_

---

> ### Author Response · Authors · 2024-11-22
>
> Dear reviewer,
> As the discussion period will soon end, we are contacting you to double-check whether our answers resolved the concerns and questions you raised during your rebuttal. Due to the cross-discipline nature of our submission, i.e., the combination of PAC-Bayesian with RL approaches, we want to ensure that any potential communication gap that seems to be reflected in your low confidence score can be overcome.
> Please let us know if we can provide further details and clarification.

---

> > ### Comment · Area_Chair_noaN · 2024-11-24
> > **Please respond to rebuttal ASAP**
> >
> > Dear reviewer,
> > The process only works if we engage in discussion. Can you please respond to the rebuttal provided by the authors ASAP?

---

> > > ### Comment · Reviewer_v8fi · 2024-11-24
> > > **Response to the authors**
> > >
> > > Thank you for your clarifications and providing additional experimental results.
> > > With regards to the result presentation, I was commenting on the final tabular results which show the final reward at the end of training which can be ambigious. And the reward curves in Figure 4, the proposed method PBAC (ours) apart from a couple of environments works on par with baselines or worse for example in the case of Hopper where there is a lot of instability.

---

> > > > ### Author Response · Authors · 2024-11-24
> > > >
> > > > Thank you for your response and additional remark.
> > > > Table 1 (a) shows the final episode reward, which we agree can be highly ambiguous. However, this is why Table 1 (b) gives the area under the learning curve over the whole training period, i.e., a scalar summary measure of the reward curves reported in Figure 4, which allows us to turn a visual qualitative evaluation into a quantitative one.
> > > > As the results provided in our general rebuttal answer show, we are performing as well as or better than the baselines in nine out of eleven experimental MuJoCo setups, as measured by pairwise one-sided t-tests to a significance level of $p < 0.05$.
> > > > Due to the common problem in RL of high inter-seed performance variance, we train each method over the same set of ten seeds. Pairwise tests allow us to use this for greater statistical power compared to a more naive unpaired t-test. Finally, we use one-sided tests as we care about whether the measured statistics perform equal or worse rather than different in general.
> > > > On the mentioned hopper environments, PBAC performs as well as or significantly better than its baselines in all three setups.
> > > >
> > > > Could the reviewer clarify which measure they prefer to evaluate the provided reward curves quantitatively?

---

### Official Review · Reviewer_1HUF · 2024-11-08

**Soundness:** 3
**Presentation:** 3
**Contribution:** 3
**Rating:** 6
**Confidence:** 3

**Summary:**

The paper addresses the challenge of efficient exploration in reinforcement learning with sparse reward tasks. This is a challenging problem that has wide reaching applications in continuous control domains. They review the literature on (Bayesian inspired) deep exploration algorithms that are designed to find solutions in this setting. They then propose a novel algorithm called PAC-Bayesian Actor critic that leverages a PAC bound to achieve efficient exploration. In practice their method involves fitting a bootstrapped ensemble of function approximators (neural network) regressed to represent the error of the one-step TD Q-learning objective. This is the first principled PAC Bayesian treatment of an Actor-Critic algorithm and it seems to perform well (albeit the evaluations are on relatively standard 'simple' control domains).

**Strengths:**

- This is a relevant problem in the field of reinforcement learning and from the reviewers limited exposure to PAC-Bayesian theory the derivation seems sound and is intuitive.
- The empirical results showcase the algorithms effectiveness particularly in tasks where existing methods for deep exploration struggle (i.e. very sparse 'ant' target finding domain, humanoid etc.) which gives a convincing picture.
- I have limited understanding of PAC-Bayesian methods but am an expert on off-policy RL. From what I can tell the derivations do indeed seem novel and the idea of modelling the error of the TD-operator with a PAC-Bayesian approach seems highly promising and worthwhile for the community.

**Weaknesses:**

- The practical implementation of the well motivated general algorithm requires a series of approximations (e.g. ensemble of networks, Gaussian assumption on the adaptive prior etc.) which are well described in the paper but strike me as somewhat complicated and ad-hoc. It would have been great to see some ablations of different choices here and provide the reader with understanding of why these choices are reasonable (I understand at least the ensemble of networks follows prior art. but the rest seems not already well established).
- The core derivation is relatively simple and clear, but the resulting method (due to the points above) becomes quite complicated. Making me wonder how easy it would be to reproduce and built on top of it.
- The empirical results still only consider fairly low-dimensional domains and no vision based policies/critics (perhaps with the exception of the humanoid domain which is fairly high-dimensional). And the domains are also mostly not naturally sparse reward problems but have been 'sparsified' from their dense reward versions. I have some concerns whether the presented method would also work in more relevant modern settings such as learning vision based policies in e.g. robotics and or more expressive function approximator classes such as using transformers or other large models. Do the authors have any intuitions on these and/or could one more relevant complex domain be considered? E.g. as in recent papers on RND etc.

**Questions:**

A discussion of the weaknesses above would be highly appreciated for me to raise my score. Additionally I have a problem in understanding one crucial part of the paper:
- The authors mention that due to the long episodes in RL a PAC-Bayesian treatment has to be done based on TD errors (and these only consider consecutive state-action-state tuples and not trajectories). This makes intuitive sense to me, however I could not fully understand how the treatment still captures the uncertainty over the full trajectory space when the estimator used for bootstrapping (i.e. of the Bellman error) conflates everything to the expectation over future trajectories. Could the authors help me understand this more clearly?

---

> ### Author Response · Authors · 2024-11-14
> **Answer to 1HUF**
>
> Thank you for your review. We will answer to each of your points below.
>
> - **The algorithm requires "a series of approximations [...] which are well described in the paper but strike me as somewhat complicated and ad-hoc**
>     Could you point to the specific approximations that seem ad-hoc? We could then offer further details and justification. In Section 3.2, the first and second steps in moving the ensemble to a function space are justified by a growing trend in the Bayesian deep-learning literature to switch from weight-space priors to function-space priors. See, e.g., Papamarkou et al. (2024) and the justifications therein. Data-informed priors (Step 3) are well-known in the PAC-Bayesian literature, and bootstrapping (Step 6) was evaluated at length by Osband et al. (2016a,1018). The remaining two (Steps 4 and 5) are indeed primarily motivated by empirical results and numerical stability. Our proposed model is not decomposable and individual effects wouldn't be visible. However, if you have a specific factor in mind that should be observed in isolation, we are happy to conduct such an experiment.
>
> - **How easy would it be to reproduce and build on top of it?**
>   We provide a full pseudo-code version of the final algorithm in Algorithm 1 (see p 23), which together with the provided hyperparameters, architectural details, and a public pytorch implementation (currently hidden to keep anonymity) of the experimental pipeline ensure reproducibility of our reported results.
>   The required critic ensemble and multi-headed actor network described in Table 6 are straightforward and the loss objective consists mainly of moment computations and their predictive mean-squared error terms, i.e., is comparable to our baselines.
>
>  - **The empirical results still only consider fairly low-dimensional domains and no vision-based policies/critics. What about more expressive function approximator classes such as using transformers or other large models?**
> Humanoid and ant both already provide a rather large observation space compared to other benchmarks (348 and 105 dimensions respectively). PBAC and the baselines share the same architectural backbone to stay as comparable as possible. Architectural advancements such as transformers, or layers adapted for spatial information, such as convolutional layers, would have to be shared with the baselines, after which we expect that the relative performance structure remains the same. Note that while the architectures are seemingly simple they already follow the latest recommendations from a recent study by Nauman et al. (2024). Of course, depending on the use case more powerful approximators, or newer ensembling methods, could directly replace the current backbone. As such the model could also be applied to vision-based benchmarks or similar. Our implementation is agnostic to such details.
> We would also like to remind, that the focus of this paper is on continuous control where the true bottleneck is the action space dimensionality rather than the observation space. Deep exploration in continuous control is far from being solved even from small action spaces. Hence, we leave scaling to future work.
>
> - **I could not fully understand how the treatment still captures the uncertainty over the full trajectory space**
> The critic network approximates the value function, which is a predictor for the discounted return of a trajectory starting from a given state and generated by following a fixed policy. Hence, an uncertainty on the value of a state given as an input directly models the uncertainty over the value whole trajectory following that state. Does this clarify the question?
>
> Please let us know in case these answers have not fully clarified your concerns and questions.
>
> _____
> Papamarkou et al., 2024: _Position: Bayesian Deep Learning is Needed in the Age of Large-Scale AI_

---

> > ### Comment · Reviewer_1HUF · 2024-12-01
> > **Thanks for the clarifications**
> >
> > I appreciate the clarifications. Regarding the open points:
> > - "How easy would it be to reproduce": It is good to know that the implementation will be open sourced.
> > - "Approximations used": As I mentioned in my review, the steps seem reasonable and justified (as you also point out again in your comment). I was merely noting that there are many steps required and they make the final objective feel a bit 'cumbersome', which makes it feel like empirical results need to be excellent to justify the approach in some sense.
> > - "Uncertainty modelling over time": No it does not clarify this properly to me. My concern is that you are bootstrapping the critic network and the bootstrap in my mind 'squashes' all uncertainty to an average value/expectation. How is uncertainty propagated through this step?
> > - "Domains": I agree that humanoid is high-dimensional but the task at hand is a sparsified version of a simple task that PPO and other algorithms easily solve and your method achieves a score that is a fraction of what is achieved in the non-sparse case (I am assuming rewards are normalized somehow?) does it constitute good behaviour? Is the humand running with a smooth gate? That aside, there are many more high-dimensional domains that aren't artificially sparsified (such as meta-world mentioned in other reviews) which would have been a good fit, the additional results provided in these domains are a good step, but the differences to REDQ here seem marginal (and probably both algorithms could benefit from some tuning).
> >
> > Overall the response is reasonable but I feel like full results on meta-world incorporated into proper paper form and with appropriate hyperparameter tuning would help a lot. I will maintain my score as is since I cannot comment further on issues raised by other reviewers.
> >
> > Overall

---

> ### Author Response · Authors · 2024-11-22
>
> Dear reviewer,
> As the discussion period will soon end, we are contacting you to double-check whether our answers resolved the concerns and questions you raised during your rebuttal. Due to the cross-discipline nature of our submission, i.e., the combination of PAC-Bayesian with RL approaches, we want to ensure that any potential communication gap that seems to be reflected in your low confidence score can be overcome.
> Please let us know if we can provide further details and clarification.

---

> > ### Comment · Area_Chair_noaN · 2024-11-24
> > **Please respond to rebuttal ASAP**
> >
> > Dear reviewer,
> > The process only works if we engage in discussion. Can you please respond to the rebuttal provided by the authors ASAP?

---

### Official Review · Reviewer_NDSb · 2024-11-12

**Soundness:** 2
**Presentation:** 2
**Contribution:** 2
**Rating:** 3
**Confidence:** 3

**Summary:**

This paper studies RL in continuous spaces, and is specifically motivated by sparse-reward problems that require exploration. They take a PAC-Bayesian perspective, and derive an algorithmic approach motivated by a PAC-Bayes analysis to address such hard exploration problems. Experimental results are given illustrating its performance on several Mujoco tasks, and that it is able to solve problems with sparse rewards.

**Strengths:**

The PAC-Bayes perspective has not been given significant attention in the RL community, and this work takes a step towards addressing that. Furthermore, the proposed algorithm does indeed appear to solve sparse-reward settings effectively.

**Weaknesses:**

1. The main theoretical result (Theorem 2) is vacuous, as is noted by the authors. It is not clear to me what purpose this result serves given this—it is trivially true that $\| Q_\pi - X \|_{P_\pi}^2 \le \frac{R^2}{(1-\gamma)^2}$, which is all this bound claims to show (unless the negative variance term can be shown to cancel out this term, but this avenue is not pursued here). Furthermore, this bound is used to motivate the algorithm, yet given that it is vacuous, one cannot draw any real conclusions from it, and thus the motivation for the algorithm becomes unclear. I would suggest removing this result and cleaning up the algorithmic motivation.
2. The experimental evaluation is limited, only considering DMC environments, and several other Mujoco tasks. Furthermore, the gains over existing methods are relatively marginal—in the majority of examples, it performs comparably to existing methods (or the stochasticity is high enough that it is not possible to give a clear ranking).
3. The writing of this paper could be improved and the overall story made more explicit. In particular, as I understand it, the main idea is to apply a PAC-Bayes analysis to get a measure of uncertainty, then use this measure of uncertainty to induce exploration in a UCB-like manner. This was not clear, however, from reading the paper. For example, there are statements like “implement the bound” (line 306) which seem to imply this but are ambiguous (what does it mean to implement a bound?). I would encourage the authors to tighten the story and make connections between the analysis and resulting algorithm more explicit. There were various other unclear statements or issues with the exposition, for example:
	* Line 266: It was not clear to me why a function is replaced by a distribution here. This may be standard in PAC-Bayes analysis (I am not too familiar with PAC-Bayes), but is not standard in the RL literature, and I would suggest further exposition here to make clear why the loss is now over a distribution.
	* Line 302-303: In the sentence starting with “However…”, what is an “existing bound rigorously developed for a specific purpose”? Some explanation of this statement (or relevant citations of such a bound) would be helpful.
	* It was difficult to parse what the final algorithm actually is. Many of the details given in Section 3.2 are not necessary for the main body, and make it difficult to determine which points are the most salient. It would be very helpful to give an algorithm box putting all the pieces together.
4. Another paper that would be good to compare against is [1].

[1] Lee, Kimin, et al. "Sunrise: A simple unified framework for ensemble learning in deep reinforcement learning." International Conference on Machine Learning. PMLR, 2021.

**Questions:**

See Weaknesses.

---

> ### Author Response · Authors · 2024-11-14
> **Answer to NDSb**
>
> Thank you for reviewing our submission and for your comments.
>
> - **W1. The main theoretical result is vacuous**.
> Indeed, the PAC-Bayes bound is, as discussed in the paper, vacuous. Its motivation is to serve as a training objective as is commonly done in PAC-Bayesian learning. Note that there is no relation between whether the bound is tight and whether it is a suitable training objective. If a non-vacuous generalization bound is desired at a later stage, the inferred posterior can be plugged into any tight PAC-Bayesian bound.
> The $\frac{nR^2}{(1 - \gamma)^2}$ which we introduce in the proof of Theorem 2 (line 851) is independent of the parameters, i.e., is irrelevant in the training process.
>
> - **W2. The experimental evaluation is limited**
>     - Regarding the limited set of results, we are currently running experiments on the Meta-World (Yu et al., 2019) benchmark, as suggested by reviewer v8fi. We will report these results as a main answer to all reviewers once they are completed.
>     - Regarding the significance. See the general answer for a pairwise one-sided t-test analysis of the results reported in Table 1. Please note that even in the currently reported way of highlighting every method whose interquartile range overlaps, we get the following ranking, which clearly shows the average improvement of PBAC upon the baselines (the brackets provide the number of overlaps with the highest IQM range).
>         - Final episode reward: DRND(1), BootDQN-P(3), BEN(4), REDQ (8), and PBAC (10)
>         - Area under learning curve: BootDQN-P (0), DRND (1), BEN (1), REDQ (5), and PBAC (8)
> - **W3. Writing of the paper and overall storyline**
>     - **_"Apply PAC-Bayes analysis [...] to induce exploration in a USB-like manner"_**
>         We essentially do posterior sampling, which is nowadays emerging as an alternative to UCB. The idea is to sample a value function from the learned posterior and act greedily with respect to it. There exist theoretical findings showing that posterior sampling is more sample-efficient than UCB (see, e.g., Tiapkin et al., 2022).
>     - **What does _"Implement the bound"_ mean?**
>     Throughout this work, we follow the common practice of designing a PAC-Bayesian bound that is subsequently used as a training objective to be minimized. The expression implementing the bound means that we derive in Section 3.2 an approximation to the theoretical bound we proved in Theorem 2. This approximation finally provides us the loss term given by equation (3) which is subsequently minimized as described in Algorithm 1.
>     - **Line 266: It was not clear to me why a function is replaced by a distribution here.**
>         We do this to capture the randomness caused by the function approximation error resulting from the approximate TD approach. Standard actor-critic methods use double critics and min-clipping to overcome the negative effects of this error. Our novelty is to model the distribution caused by this error and use the resulting uncertainty for deep exploration.
>     - **It was difficult to parse what the final algorithm actually is.**
>         We described the final algorithm in Algorithm 1 in the appendix (see p23) which implements equation (3). We aim to explain every step in Section 3.2. Could you tell which of these steps is unclear or not sufficiently justified? Does the algorithm make the specific steps clearer, or would a further explanation be helpful?
>
> - **W4. Another paper that would be good to compare against is Sunrise.**
> Thank you for the reference, we will add it to our related work section. SUNRISE (Lee et al., 2021) uses an ensemble of actor critic nets in a SAC setup and weights the Bellman error by the empirical standard deviation of their predictions. Additionally, they rely on a bootstrapping mask, as do we, and perform UCB-based exploration. Their target can be interpreted as roughly corresponding to a weighted version of the first term in our objective in Equation (3).
>
> Please let us know whether these answers clarify your questions or whether you have further requests or additional questions.
>
> _____
> Lee et al., 2021: _SUNRISE: A Simple Unified Framework for Ensemble Learning in Deep Reinforcement Learning_
> Tiapkin et al., 2022: _Optimistic posterior sampling for reinforcement learning with few samples and tight guarantees_Optimistic posterior sampling for reinforcement learning with few samples and tight guarantees_

---

> ### Author Response · Authors · 2024-11-22
>
> Dear reviewer,
> As the discussion period will soon end, we are contacting you to double-check whether our answers resolved the concerns and questions you raised during your rebuttal. Due to the cross-discipline nature of our submission, i.e., the combination of PAC-Bayesian with RL approaches, we want to ensure that any potential communication gap that seems to be reflected in your low confidence score can be overcome.
> Please let us know if we can provide further details and clarification.

---

> > ### Comment · Reviewer_NDSb · 2024-11-22
> > **Response**
> >
> > Thanks to the authors for their answers and the additional experiments on Metaworld.
> >
> > Regarding the vacuous bound, I still find this problematic. It is true that $\mathbb{E}||Q_\pi - X||\_{P\_\pi}^2 \le \frac{R^2}
> > {(1-\gamma)^4}$, so it is also trivially true that $\mathbb{E}||Q\_\pi - X||\_{P\_\pi}^2 \le \frac{R^2}
> > {(1-\gamma)^4} + C$ for any $C \ge 0$. Therefore, I can simply choose $C$ to be anything I want, in particular I could set $C = \hat{L}\_{\mathcal{D}}(\rho)$, or any other function of $\rho$. This does not imply, though, that minimizing this term with respect to $\rho$ is the correct thing to do---if I choose $C$ to depend on $\rho$ arbitrarily this would clearly not be the case. Given this, I still believe the paper needs to be revised to either tighten the bound to avoid the $\frac{R^2}
> > {(1-\gamma)^4}$ dependence, or the motivation for the algorithm changed so that it does not rely on this result.

---

> ### Author Response · Authors · 2024-11-24
>
> We agree with the difficulties the vacuousness of the bound creates in interpreting our contribution. It is easily fixable by minor changes in the manuscript following the recipe below.
>
> The vacuousness stems from the worst-case upper bound we build on the term:
>
> $$
> \log \Big( \frac{1}{\delta} E_{D\sim P_D} E_{h\sim \rho_0} e^{nd(\hat L_D(h),L(h))}\Big)
> $$
>
> in Line 848 of the proof of Theorem 2. Define $\lambda_t \in [0,1]$ as the operator norm of the state transition kernel $P_{\pi_t}(s_{t+1}|s_t, \pi_t(s_t))$ employed during the construction of the replay buffer at environment interaction time $t$. Then defining a non-stationary Markov chain on the sample Bellman operator error $(\tilde T_{\pi_t} Q(s_t,a_t,s_{t+1}) - Q(s_t, a_t))$ and applying Theorem 5 of Fan et al. (2021) with a trivial union bound argument on two alternative orders of the random variable and its expectation (as standard in the Hoeffding inequality variants) as well as few small algebraic manipulations we attain
>
> $$
> \log \Big (  E_{D\sim P_D} E_{h \sim \rho_0} e^{n (L(h)-\hat L_D(h))} \Big ) \leq \frac{(1+\max_{t \in [n]}  \lambda_t)R}{(1-\max_{t \in [n]}  \lambda_t) 8}.
> $$
>
> Plugging this result into Line 848 of our submission with further few small steps yields the following inequality
>
>
> $E_{X \sim \rho} ||Q_\pi - X||^2_{P_\pi} \leq
>  \frac{1}{(1-\gamma)^2} \Bigg ( \hat L_D(\rho) + \frac{1}{n} \Big [ KL(\rho || \rho_0) + \log \frac{1}{\delta} + \frac{(1+\max_{t \in [n]} \lambda_t)R}{(1-\max_{t \in [n]} \lambda_t) 8} \Big ]   \Bigg )- \gamma^2 E_{X \sim \rho} var_{s \sim P_\pi}[X(s,\pi(s))]$.
>
> We will replace it with Eq. (2) of the current version of our manuscript. Although calculating $\lambda_t$'s is not practical for many MDPs, the term it appears in the bound does not depend on the posterior distribution, hence it boils down to constant in the policy evaluation step. Hence, all the rest of our argumentation, algorithm, and experiment results remain valid.
>
> Does this address your concern?
>
> ____
> Fan et al., 2021: _Hoeffding's Inequality for General Markov Chains and Its Applications to Statistical Learning_, JMLR, 2021

---

> > ### Comment · Reviewer_NDSb · 2024-11-24
> > **Response**
> >
> > Thanks to the authors for the follow-up. I have two concerns about this argument:
> >
> > 1.  The maximum eigenvalue of a transition matrix is 1, so I believe $\max_{t \in [n]} \lambda_t = 1$, which causes this new term to blow up.
> >
> > 2. In addition, I believe $L(h) - \hat{L}_D(h)$ should behave as $O(\sqrt{n})$, by standard concentration arguments, but it appears here to scale as $O(1)$. This would cause the resulting term to go down as $O(1/\sqrt{n})$ rather than $O(1/n)$ as stated here (which does lead to a non-vacuous bound, though this term may dominate the $\hat{L}_D(\rho)$ term).
> >
> > Could the authors comment on this?

---

> > > ### Author Response · Authors · 2024-11-25
> > >
> > > - Yes, blows up if the spectral gap of the transition kernel is 1, which means the chain keeps perfect autocorrelation no matter how long it is. This is indeed a risk for the worst-case scenario, but is not realistic. Furthermore, this is an epistemic limit any similar bound statement has to live with.
> > >
> > > - We believe the reviewer refers to McAllester-type bounds where the complexity term appears in a square root and this term has the data size in its denominator. This result follows from choosing the distance between expected and the true risk, the $d(\cdot,\cdot)$ function in our notation, and applying the Pinsker's inequality afterwards. It is indeed possible to follow the same path in our bound and obtain a similar form, as the solution we provided above applies to the LogSumExp term, which is used in a different step in the bound proof. We instead followed a version akin to Catoni-type bounds that has a free parameter to choose, which compensates for the missing square-root term. In the bound given in our previous comment, we chose it to be 1/n. With other options it may be made tighter. We are able to easily provide both versions in the final version of the paper, if requested.

---

### Author Response · Authors · 2024-11-14
**General Answer**

We thank all reviewers for reading our submission in detail. This answer contains shared answers, with further answers given individually to each reviewer. To assure that our clarifications are sufficient we will provide them first here and will subsequently update the pdf throughout the discussion period whenever the reviewers agree.

- We will provide additional results on Meta-World (Yu et al., 2019) as suggested v8fi. The experiments are currently running and we will update this answer whenever they are ready. This provides us with a second set of experiments, extending the results we have reported so far.
- Regarding the significance of the current results. The following two tables provide the results of pairwise one-sided t-tests between the highest mean and the others, with the null hypothesis being that two means are equal. Cells marked with _X_ indicate that the hypothesis was not rejected (for $p \leq 0.05$ ). Ranking the method by their performance, we have that overall PBAC performs as well as or better than the other models in most environments.
    - Final reward: DRND (3), BootDQN-P (4), BEN (5), REDQ (8), PBAC (9)
    - Area under learning curve: BootDQN-P (0), BEN (1), DRND (2), REDQ (6), PBAC (9)

### Final episode reward
| Environment | BEN | BootDQN-P | DRND | REDQ | PBAC (ours) |
| -------- | -------- | -------- |-------- | -------- | -------- |
| ant | | | | X | X |
| hopper | | X | X | X | X |
| humanoid | | | X | X | |
| ballincup | X | X| X | X | X |
| cartpole | | | | | X|
| reacher| X | | | X | |
| ant (sparse) | | | | X | X|
| ant (very sparse) | X | | | | X |
| hopper (sparse) | X | X | | X | X |
| hopper (very sparse) | X | X |  | X | X |
| humanoid (sparse) | | | | | X |


### Area under learning curve
| Environment | BEN | BootDQN-P | DRND | REDQ | PBAC (ours) |
| -------- | -------- | -------- |-------- | -------- | -------- |
| ant | | | | X | X |
| hopper | | | | X | X |
| humanoid | | | X | X | |
| ballincup | X | | | X | X |
| cartpole | | | | | X |
| reacher| | | | X | |
| ant (sparse) || | | | X |
| ant (very sparse) | | | | | X |
| hopper (sparse) | | | | | X |
| hopper (very sparse) | | | | X | X |
| humanoid (sparse) | | | | | X |

_____
Yu et al., 2019: _Meta-World: A Benchmark and Evaluation for Multi-Task and Meta Reinforcement Learning_

---

> ### Author Response · Authors · 2024-11-19
> **Preliminary results on Meta-World (Yu et al., 2019)**
>
> We provide a first set of results on five sparse environments, comparing PBAC with REDQ, our strongest baseline in the original submission. We follow Fu et al. (2024)'s sparse experimental setup, where an agent only receives a reward upon task completion, e.g., when it successfully opens a window, without any intermediary reward.
> As our computational resources are limited during this rebuttal, we consider a task to be solved after the agent has received a success rate of 100% for two consecutive evaluation episodes. We will provide success curves over a longer run of 1e6 environmental steps in the camera-ready version.
> We report the average number of steps needed over three seeds. The `*` indicates that PBAC failed in one of the seeds and that the reported average is over the remaining two.
>
>
> | Name                | PBAC   | REDQ   |
> |---------------------|--------|--------|
> | `drawer_open_goal_hidden`  | 280000*   | **193333** |
> | `drawer_close_goal_hidden` | **33333**  | 46666  |
> | `window_open_goal_hidden`  | **73333**  | 100000 |
> | `window_close_goal_hidden` | **60000**  | 93333  |
> | `reach_goal_hidden`        | **106666** | 160000 |
>
>
> In four out of the five tasks, PBAC is clearly better than REDQ. Only on `drawer_open_goal_hidden` does it perform worse.
> Note that these results are without hyperparameter tuning, i.e., using the default parameters we reported in our paper. We expect these numbers to improve further with tuned hyperparameters.
>
> The remaining three baselines reported in our paper are currently training and we will update the table once they are ready. Preliminary results indicate that all three perform worse.
>
> _____
> Fu et al., 2024: _FuRL: Visual-Language Models as Fuzzy Rewards for Reinforcement Learning_
> Yu et al., 2019: _Meta-World: A Benchmark and Evaluation for Multi-Task and Meta Reinforcement Learning_

---

> > ### Author Response · Authors · 2024-11-21
> > **Additional results on Meta-World**
> >
> > We now have the final results for the remaining baselines.  As we can see the additional baselines struggle a lot in most experimental setups. Please let us know if further experiments are required.
> >
> > | Name                | PBAC   | REDQ   | BEN | DRND | BootDQN-Prior |
> > |---------------------|--------|--------| ----- | ----- | ----- |
> > | `drawer_open_goal_hidden`  | 280000*   | **193333** | _Fail_ | 440000 | _Fail_ |
> > | `drawer_close_goal_hidden` | **33333**  | 46666  | **33333** | 40000 | 113333 |
> > | `window_open_goal_hidden`  | **73333**  | 100000 | _Fail_ | 90000* | 280000** |
> > | `window_close_goal_hidden` | **60000**  | 93333  | _Fail_ | 306666 | 420000 ** |
> > | `reach_goal_hidden`        | **106666** | 160000 | 90000* | 40000* | _Fail_ |
> >
> > All methods are run on three seeds. _Fail_ refers to all three seeds failing, `*` and `**` indicate that the method failed in one or two seeds respectively and the average is computed over the remaining seeds. The lowest average over three seeds is marked **bold** in each row. As we can see the additional baselines struggle a lot in most experimental setups.

---

### Public Comment · ~Haque_Ishfaq1 · 2024-11-20
**Missing related work on randomized exploration method**

Dear authors,
In section 2.2, where you discuss randomized exploration based methods, you missed some key recent works.

In Section 2.2 (ii), where you discuss randomized value iteration based method, you should also discuss Ishfaq et al 2021 which proposes RLSVI-PHE algorithm with provable regret guarantee under general function approximation setting.

In Section 2.2 (iii), where you discusss policy randomization via network parameter perturbation, you should discuss Ishfaq et al 2024a and Ishfaq et al 2024b. Both these papers proposed Langevin Monte Carlo and underdamped Langevin Monte Carlo based Thompson sampling algorithm. They also perform perturbation in the Q network parameter using Langevin update.

I would appreciate if you could cite these works as they are highly relevant and SOTA randomized exploration methods with provable regret guarantees.

```
Ishfaq, Haque, et al. "Randomized exploration in reinforcement learning with general value function approximation." International Conference on Machine Learning. PMLR, 2021.

Ishfaq, Haque, et al. "Provable and Practical: Efficient Exploration in Reinforcement Learning via Langevin Monte Carlo." The Twelfth International Conference on Learning Representations. (2024a)

Ishfaq, Haque, et al. "More Efficient Randomized Exploration for Reinforcement Learning via Approximate Sampling." Reinforcement Learning Conference (2024b).

```

---

### Meta-Review · Area_Chair_noaN · 2024-12-20

**Metareview:**

This paper derives a deep exploration method from the perspective of PAC-Bayes. They treat an ensemble as an empirical estimate of a posterior, and apply posterior sampling during training time for exploration and Bayesian model averaging during evaluation time. They show results on a number of continuous control tasks.

Strengths:
The problem of deep exploration is important and the authors have an interesting take on it
The method does seem to have better performance on some tasks
The PAC-Bayes approach is underappreciated

Weaknesses:
The bound as noted by the authors themselves is vacuous, and the justifications with Reviewer NDSb were not very convincing
The simplification from the insight to the actual practical method is pretty hard to parse and non-intuitive, as noted by reviewers
The results on the benchmarks are not that compelling, REDQ does very comparably.
The paper could benefit from some additional analysis of the results.

The vacuous bound and relatively small gains over baselines prompt me to recommend paper rejection, but the work has high potential in the future!

**Additional Comments On Reviewer Discussion:**

The reviewers brought up great points about empirical results and a vacuous bound, as well as readability and experimental analysis questions. Reviewer NDSb in particular is knowledgeable about the theoretical underpinnings of this work and was not convinced by either the theory or empirical results. This along with the assessment of 1HUF about readability and empirical results motivates a rejection for this paper in it's current form.

---

### Decision · Program_Chairs · 2025-01-22

Reject